# Genome-wide association and multi-omic analyses reveal *ACTN2* as a gene linked to heart failure

Marios Arvanitis[1,2], Emmanouil Tampakakis[2], Yanxiao Zhang [3], Wei Wang[4], Adam Auton[4], 23andMe Research Team*, Diptavo Dutta [1,5], Stephanie Glavaris [2], Ali Keramati [2], Nilanjan Chatterjee[5,6], Neil C. Chi [7,8], Bing Ren [3,8], Wendy S. Post[2,9] & Alexis Battle [1✉]

Heart failure is a major public health problem affecting over 23 million people worldwide. In this study, we present the results of a large scale meta-analysis of heart failure GWAS and replication in a comparable sized cohort to identify one known and two novel loci associated with heart failure. Heart failure sub-phenotyping shows that a new locus in chromosome 1 is associated with left ventricular adverse remodeling and clinical heart failure, in response to different initial cardiac muscle insults. Functional characterization and fine-mapping of that locus reveal a putative causal variant in a cardiac muscle specific regulatory region activated during cardiomyocyte differentiation that binds to the *ACTN2* gene, a crucial structural protein inside the cardiac sarcolemma (Hi-C interaction *p*-value = 0.00002). Genome-editing in human embryonic stem cell-derived cardiomyocytes confirms the influence of the identified regulatory region in the expression of *ACTN2*. Our findings extend our understanding of biological mechanisms underlying heart failure.

[1] Department of Biomedical Engineering, Johns Hopkins University, Baltimore, MD, USA. [2] Department of Medicine, Division of Cardiology, Johns Hopkins University, Baltimore, MD, USA. [3] Ludwig Institute for Cancer Research, San Diego, CA, USA. [4] 23andMe, Inc., Mountain View, CA, USA. [5] Department of Biostatistics, Johns Hopkins Bloomberg School of Public Health, Baltimore, MD, USA. [6] Department of Oncology, School of Medicine, Johns Hopkins University, Baltimore, MD, USA. [7] Department of Medicine, Division of Cardiology, University of California, San Diego, La Jolla, CA 92093, USA. [8] School of Medicine, Institute of Genomic Medicine, University of California, San Diego, La Jolla, CA 92093, USA. [9] Department of Epidemiology, Johns Hopkins Bloomberg School of Public Health, Baltimore, MD, USA. *A list of members and their affiliations are listed at the end of the paper. ✉email: ajbattle@jhu.edu

Heart failure is a highly prevalent disease[1] that constitutes a major medical and economic burden in the healthcare system, accounting for ~1–2% of the annual healthcare budget in developed countries[2]. Although almost any disease that directly or indirectly affects myocardial function can lead to the eventual development of clinical heart failure, it is well-established that certain intrinsic homeostatic mechanisms like the renin–angiotensin–aldosterone axis and the sympathetic nervous system potentiate the effects of a variety of myocardial insults and cause adverse left ventricular remodeling[3], suggesting that multiple cellular mechanisms that lead to the disease are shared regardless of the inciting condition.

The increasing appreciation of an underlying strong heritable component of clinical heart failure further strengthens the argument for shared, yet unidentified, disease mechanisms whose discovery could reveal novel targets for its treatment and prevention. Indeed, large recent pedigree studies estimate heart failure heritability to be 26–34%[4]. However, large-scale genome-wide associations studies (GWAS) for heart failure have been unsuccessful to-date at uncovering a significant proportion of this estimated heritability underscoring a major unmet need in cardiovascular genetics. In fact, the largest published GWAS for heart failure until recently had only identified one genome-wide significant locus for all-comers with the disease that the investigators attribute to its overlap with atrial fibrillation[5]. A larger GWAS performed and published in parallel to our study increased the number of identified loci to 11[6]. Even within this important work, however, many of the identified loci appear to be acting via heart failure risk factors and these loci have not yet been extensively functionally characterized, thereby limiting identification of actionable targets that predispose to heart failure development.

In the current work, we perform a large-scale GWAS for heart failure and replicate our findings in a comparably sized independent cohort. We identify and replicate associations between heart failure and one known locus in chromosome 4 near the *PITX2* gene and two novel loci near the *ABO* (chromosome 9) and *ACTN2* (chromosome 1) genes. One of the novel loci near *ABO* was also detected in the aforementioned recently published GWAS[6]. Heart failure sub-phenotyping and multi-trait conditional analyses show that the novel chromosome 1 locus affects heart failure and left ventricular remodeling independently of known risk factors and in response to a variety of initial cardiac muscle insults. Detailed functional characterization of that locus using epigenomic, Hi-C, and transcriptomic datasets in differentiating cardiomyocytes reveals a cardiac muscle-specific regulatory element that is dynamic during cardiomyocyte differentiation and binds to the promoter of the *ACTN2* gene, whereas genome-editing confirms that *ACTN2* expression is significantly reduced in cardiomyocytes that carry a deletion of the identified novel regulatory element.

## Results and discussion

**GWAS meta-analysis identifies novel heart failure loci.** We performed a large-scale GWAS meta-analysis of five cohorts that study cardiovascular disease and two population genetics cohorts, all of European ancestry comprising a total of 10,976 heart failure cases and 437,573 controls. We used the 1000 Genomes phase 3 reference panel to impute variants from single nucleotide polymorphism (SNP) array data and analyzed a total of 13,066,955 unique genotyped or high-confidence imputed variants (INFO score > 0.7) with a minor allele frequency >1%. We analyzed each individual cohort using a logistic mixed model and meta-analyzed all studies with fixed effects inverse-variance meta-analysis.

The combined meta-analysis revealed one previously identified and two novel loci associated with clinical heart failure at a genome-wide significance threshold (p-value < 5e-8) (Fig. 1a, Table 1, and Supplementary Data 1). All identified leading variants are common (MAF > 10%) and are located in non-coding regions of the genome (Supplementary Fig. 1). We validated our genome-wide significant loci in an independent cohort of 24,829 self-reported heart failure cases and 1,614,513 controls of European ancestry from the personal genetics company 23andMe, Inc. with all three sentinel variant associations successfully replicating at a nominal p-value level (p < 0.05) (Table 1) and after Bonferroni adjustment. Demographic information comparing the Discovery and Replication cohorts is available in Supplementary Table 1.

**Analysis links heart failure and musculoskeletal traits.** We subsequently performed linkage disequilibrium (LD) score regression to estimate heart failure heritability driven by common variants and the genetic correlation between heart failure and other complex traits. Liability scale SNP heritability for the disease assuming a population prevalence of 1.8%[7] was 5.9% (SE 0.7%), much lower than the pedigree-based estimates of 26%, a discrepancy that has been observed for other complex traits[8] and could be explained by multiple factors, including rare variants. The LD score regression intercept was 0.99, indicating no inflation beyond what can be accounted for by polygenicity. As expected, we saw significant genetic correlation between heart failure and known heart failure risk factors, such as hypertension, ischemic heart disease, adverse lipid profiles, diabetes, and atrial fibrillation. We also found a strong association between heart failure and pulmonary, musculoskeletal, and GI traits (Fig. 1b and Supplementary Table 2). We should note that genetic correlation analysis should not be viewed as evidence for a causal relationship between the tested diseases and consequently these results do not indicate that heart failure is causally influenced by musculoskeletal disorders. However, these disorders may share some genetic factors or cellular pathways—since the heart is composed mostly of muscle and stromal tissue, it is plausible that it could share regulatory mechanisms with other organs of similar cell type composition.

**Atrial fibrillation's role in heart failure development.** We investigated each genome-wide significant locus in depth. The chromosome 4 locus tagged by the SNP rs1906615 is found in an intergenic region close to the *PITX2* gene. This locus was previously identified as containing the strongest evidence for association with atrial fibrillation[9] and has been reported as a significant locus in recent heart failure GWAS[5,6]. However, that association was thought to be mediated via the relative enrichment of the heart failure population in atrial fibrillation cases[5]. Indeed, via multi-trait conditional and joint analysis using summary statistics from GWAS of atrial fibrillation, we confirm that the effect of the *PITX2* locus on heart failure is explained by its effect in atrial fibrillation (Table 2 and Supplementary Data 2). Mendelian randomization (MR) analysis using 110 independent (LD $r^2 < 0.001$) genome-wide significant atrial fibrillation-associated variants, provides further evidence for a directional effect of atrial fibrillation on heart failure development (weighted mode MR effect size 0.21 (odds ratio 1.23), z-test p < 0.0001). Sensitivity analysis using the MR Egger and weighted median approach to account for potential pleiotropy and/or invalid instruments confounding our MR estimates is statistically significant and supports our hypothesis for a causal effect of atrial fibrillation on heart failure (Supplementary Fig. 2). While MR methods alone

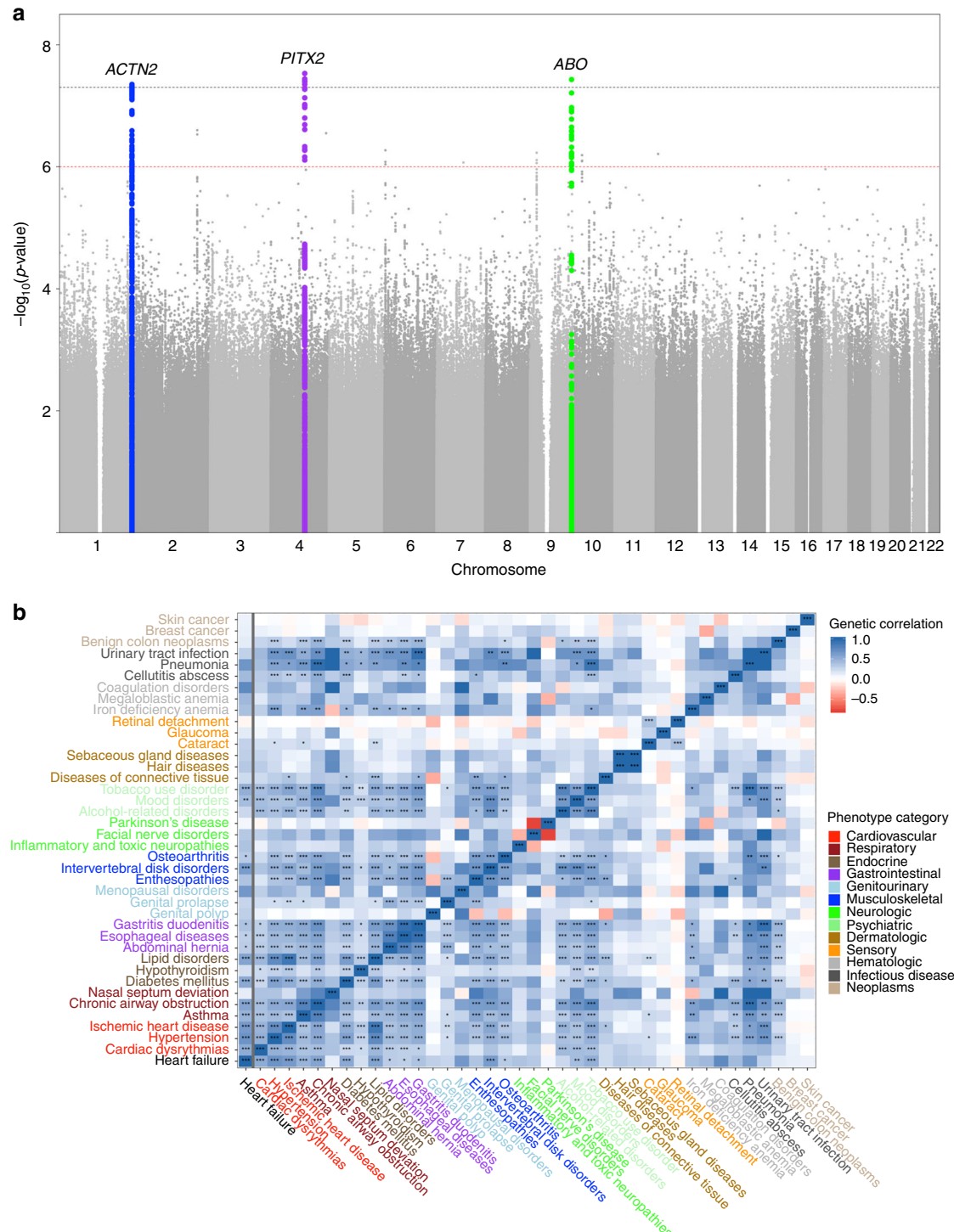

**Fig. 1 Summary GWAS and genetic correlation plots. a** Manhattan plot of the GWAS meta-analysis. The black dotted line denotes the genome-wide significance threshold (p-value < 5e-8) while the red dotted line denotes the suggestive threshold (p-value < 1e-6). N = 448,549 independent participants. **b** Genetic correlation values between heart failure and other complex traits. The x-axis shows the genetic correlation r-value. Traits are color coded based on the group in which they belong. Stars denote the Bonferroni adjusted z-test p-value: (*) 0.001 < p-value < 0.01, (**) 0.0001 < p-value < 0.001, (***) p-value < 0.0001. N = 1,171,562 SNPs. Source data are provided as a Source Data file.

cannot rule out reverse causation, 105/110 of the variants used here have a larger effect size for atrial fibrillation than heart failure, and atrial fibrillation displays greater SNP heritability, supporting that the MR result indicates an effect on Heart Failure that is mediated through atrial fibrillation rather than the reverse.

**ACTN2 gene enhancer is associated with heart failure**. The chromosome 1 locus tagged by the SNP rs580698 is found near *ACTN2*, a gene that encodes for a structural cardiac protein inside the sarcolemma, at which rare mutations have recently been associated with the development of cardiomyopathy and consequently heart failure[10]. Multi-trait conditional and joint analysis

**Table 1 Lead variant associations in the Discovery and Replication cohorts.**

| Sentinel variant | Effect allele | Discovery effect size | Discovery standard error | Discovery p-value[a] | Discovery min imputation r² | Replication effect size | Replication standard error | Replication p-value[a] | Replication min imputation r² |
|---|---|---|---|---|---|---|---|---|---|
| rs580698 | A | 0.13 | 0.02 | 4.5e-08 | 0.988 | 0.04 | 0.01 | 0.011 | 0.996 |
| rs1906615 | T | 0.10 | 0.02 | 2.9e-08 | 0.958 | 0.05 | 0.01 | 3.1e-05 | 0.994 |
| rs9411378 | A | 0.11 | 0.02 | 3.7e-08 | 0.897 | 0.05 | 0.01 | 4e-07 | 0.957 |

[a]Score test p-value.

**Table 2 Lead GWAS variants in multi-trait analysis, Heart Failure sub-phenotypes and echocardiographic traits.**

| Sentinel variant | All cause heart failure (N = 448,549) | Multi-trait conditional analysis[a] (N = 448,549) | Ischemic heart failure (N = 27,068) | Non-ischemic heart failure (N = 27,068) | HFrEF (N = 18,498) | HFpEF (N = 18,498) | LVEF (N = 18,498) | LVEDD (N = 9150) | IVSD (N = 9577) |
|---|---|---|---|---|---|---|---|---|---|
| rs580698 | 4.5e-08 (0.13) | 6.7e-08 (0.13) | 0.02 (0.15) | 0.005 (0.16) | 0.07 (0.18) | 0.81 (0.02) | 0.20 (−0.10) | 0.05 (0.20) | 0.39 (0.02) |
| rs1906615 | 2.9e-08 (0.10) | 0.005 (0.05) | 0.58 (−0.03) | 0.03 (0.10) | 0.08 (−0.13) | 0.14 (0.09) | 0.69 (0.02) | 0.51 (−0.05) | 0.23 (0.03) |
| rs9411378 | 3.7e-08 (0.11) | 1.4e-07 (0.11) | 0.24 (0.06) | 0.13 (0.08) | 0.36 (0.07) | 0.26 (0.07) | 0.23 (0.08) | 0.10 (0.17) | 0.44 (−0.02) |

All cell values are formatted as score test p-value (effect size estimate). N denotes the total sample size for each analysis.
HFpEF heart failure with preserved ejection fraction, HFrEF heart failure with reduced ejection fraction, IVSD interventricular septum diameter, LVEDD left ventricular end diastolic diameter, LVEF left ventricular ejection fraction.
[a]Multi-trait analysis is conditioned on the following heart failure risk factors: ischemic heart disease, hypertension, diabetes mellitus, atrial fibrillation.

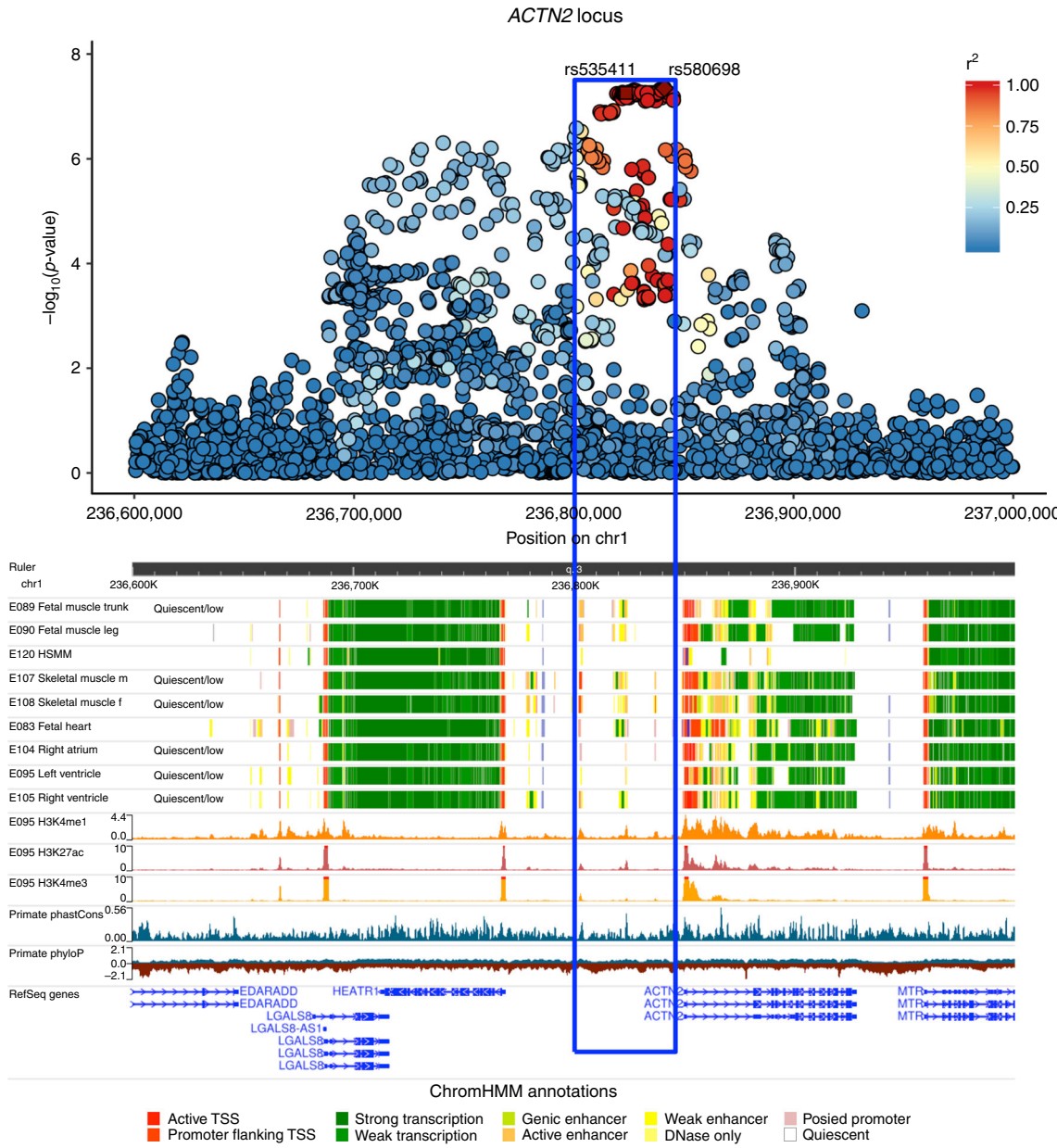

**Fig. 2 Epigenetic overview of the *ACTN2* locus.** Manhattan plot of the *ACTN2* locus and the corresponding Roadmap ChromHMM 25-state model annotations in cardiac and muscle cell-types and tissues, H3K4me1, H3K4me3, H3K27ac peaks in left ventricle, and phyloP and phastCons evolutionary conservation values. The blue box highlights the region that contains all 111 credible set variants identified by CAVIAR.

with common heart failure risk factors (atrial fibrillation, ischemic heart disease, hypertension, diabetes mellitus) does not result in a significant change in the effect of the *ACTN2* locus on heart failure, suggesting that the association signal is not primarily mediated via these other diseases (Table 2 and Supplementary Data 2). A phenome-wide association approach (PheWAS) using echocardiographic and other phenotypic information available for a subset of our cohorts and participants demonstrates that the *ACTN2* locus is significantly associated with both ischemic and non-ischemic heart failure and has a trend for an effect in left ventricular dilation and heart failure with reduced ejection fraction, thereby suggesting its potential role in mechanisms predisposing to left ventricular adverse remodeling in response to various initial insults (Table 2 and Supplementary Fig. 3). Chromatin state data for the *ACTN2* locus from Roadmap Epigenomics reveal a broad area of muscle-specific active enhancer elements in the skeletal muscle, fetal

heart, left and right ventricular tissues (Figs. 2 and 3a, and Supplementary Fig. 4). Integration with expression quantitative trait loci (eQTL) data does not reveal any compelling evidence of colocalization between the GWAS signal and altered expression of nearby genes in adult blood or post-mortem adult heart tissues (Supplementary Table 3). In addition, no significant association with the expression of nearby genes is detected in eQTL studies performed with freshly preserved heart tissue at the time of heart transplant/donor heart explant[11].

Since eQTL analysis in adult tissue did not identify a target gene for the locus, in an effort to provide a credible hypothesis of how this locus is associated with heart failure, we proceeded with additional functional characterization. The first step was to fine map putative causal variants within the locus. First, we generated a credible set of SNPs for the *ACTN2* locus based on the GWAS associations and linkage disequilibrium pattern for each region of interest using CAVIAR[12] and selected the 111 SNPs with GWAS

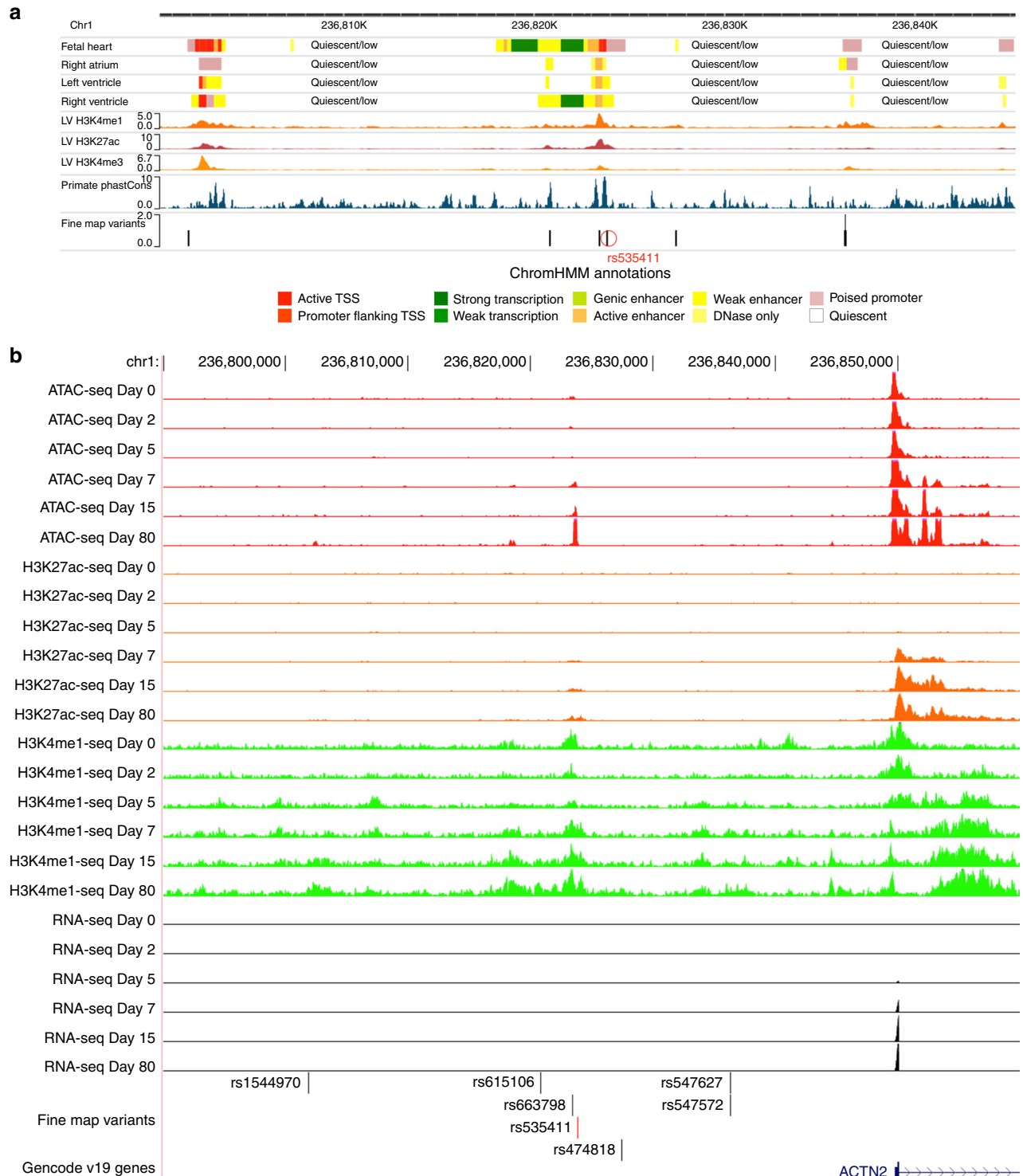

**Fig. 3 Fine-mapping of the *ACTN2* locus. a** A zoomed in view of the credible set region for the *ACTN2* locus in Roadmap Epigenomics. **b** ATAC-seq, ChIP-seq (H3K27ac, H3K4me1), and RNA-seq peaks in the hESC-to-cardiomyocyte differentiation model for the fine-mapped SNPs that overlap cardiac-specific active chromatin states. We see that one SNP (rs535411) is overlapping an ATAC-seq peak that starts to develop on Day 7 of cardiomyocyte differentiation and remains active until Day 80. Ranges for peak values: ATAC-seq: 0–127, H3K27ac-seq: 0–127, H3K4me1-seq:0–10. **b** was created using the UCSC genome browser (http://genome.ucsc.edu/).

*p*-value < 5e-7 from the credible set. Then, we intersected that set of SNPs with active chromatin states using the Roadmap Epigenomics ChromHMM 25-state model[13] for cardiac tissues and with candidate cis-regulatory elements (ccREs) from the ENCODE registry[14]. Of 111 strongly associated variants in almost perfect LD within the credible set for *ACTN2*, only seven

overlapped regulatory elements in both Roadmap and ENCODE and were, therefore, used in downstream analyses (Supplementary Data 3).

Next, we verified the presence of active chromatin states overlapping our *ACTN2* locus SNPs in engineered human embryonic stem cell (hESC) derived cardiomyocytes during

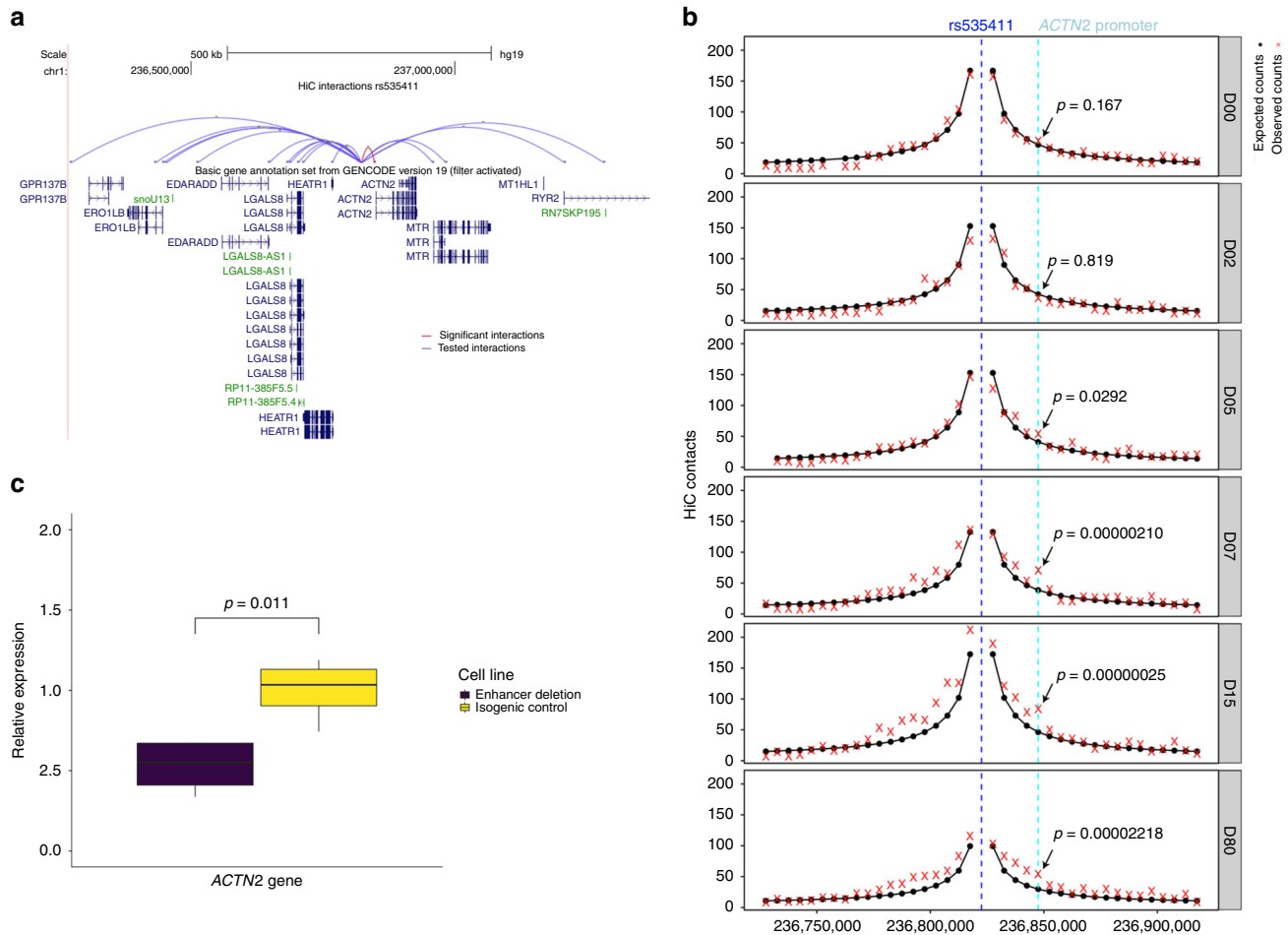

**Fig. 4 The fine-mapped regulatory element in chromosome 1 affects *ACTN2* gene expression. a** HiC data from cardiomyocytes on Day 80 of differentiation for the interaction between the 5 Kb region containing rs535411 and the promoters of nearby (within 1 Mb) genes. Significant interactions after Bonferroni adjustment are colored red, while non-significant interactions (Bonferroni Poisson test *p* > 0.05, *n* = 2 independent biological replicates) are colored blue. **b** HiC data in different stages of cardiomyocyte differentiation centered at rs535411. Black lines/dots: expected interaction counts, which follow distance-based decay, red crosses: observed interaction counts, Blue dashed line: 5 Kb region containing rs535411, cyan dashed line: 5 Kb region containing *ACTN2* promoter, *p*-value indicates the upper-tailed Poisson probability with the expected counts as lambda. *N* = 2 independent biological replicates. **c** Boxplot of relative expression of the *ACTN2* gene in cardiomyocytes edited to produce a deletion of the identified putative enhancer element and isogenic controls (*n* = 4 independent biological replicates). We see that expression of *ACTN2* is reduced by 47% in the cardiomyocytes carrying the deletion. Boxplot center line represents the median, the bounds represent the interquartile range (IQR) (25–75%) and the whiskers extend from the bounds to the largest value no further than 1.5*IQR from the bound. Data beyond the end of the whiskers are plotted individually. Source data are provided as a Source Data file.

different stages of differentiation. We showed that one of the seven target variants, rs535411 (Supplementary Data 4) overlaps cardiomyocyte-specific ATAC-seq, H3K4me1, and H3K27ac peaks that start to appear on day 7 of hESC differentiation into a cardiomyocyte and persist until at least day 80 (Fig. 3b). The ATAC-seq signal onset at that region coincides temporally with the onset of ACTN2 expression based on RNA-seq data from the same differentiation experiment (Fig. 3b), with both occurring between day 5 and 7. High-resolution chromatin conformation capture (HiC) analysis of our hESC to cardiomyocyte model on day 80 of differentiation shows that the ATAC-seq peak is in contact with the *ACTN2* gene promoter (observed/expected interaction frequency = 1.82, Poisson test *p* = 0.00002) (Fig. 4a and Supplementary Table 4) and its interaction is dynamic and increases during differentiation (Fig. 4b).

Although rare variants within the *ACTN2* gene are known to be associated with cardiomyopathies, the credible set analysis does not support the coding region as being the primary driver of the GWAS signal. Moreover, conditioning on the sentinel variant

eliminates the signal for association of the locus with heart failure, suggesting that the association is driven primarily by a single causal variant in high LD with the sentinel SNP (Supplementary Fig. 5). We should note however that we cannot exclude the possibility that the association signal could be caused or increased by other rare variants within our identified cardiac muscle enhancer region that contains rs535411.

We subsequently ventured to experimentally validate the effect of the putative enhancer element at the 1q43 locus on *ACTN2* gene expression in cardiomyocytes. For that purpose, we generated engineered hESCs with a CRISPR-Cas9-induced deletion in the ~2200 bp region that delimits the enhancer element identified in our hESC-CM epigenomic data analyses. We differentiated these edited hESCs into cardiomyocytes and on day 15 of differentiation, we compared the expression of *ACTN2* to that of isogenic hESC-CMs without the deletion. *ACTN2* expression was reduced on average by half in the edited hESC-CMs compared to controls (Fig. 4c). We then assessed expression of other nearby genes, and none appeared to be affected by the

deletion (Supplementary Fig. 6A). These experiments support the epigenetic predictions of a cardiac enhancer element in that region and validate the Hi-C data that suggest binding of that enhancer element to the *ACTN2* gene promoter. More importantly, these results provide a mechanistic hypothesis of the GWAS association between the *ACTN2* locus and heart failure. Indeed, previous studies have established that reduction of *ACTN2* mRNA levels via a siRNA leads to defects in the number and size of cardiac sarcomeres along with a phenotype of dilated heart with thin walls and a decreased heart rate in zebrafish[15]. It is therefore plausible that smaller reductions of *ACTN2* expression as those caused by variants within our identified enhancer could generate subtler cardiac sarcomeric defects in humans that become apparent later in life in individuals with an additional genetic or environmental insult to the heart muscle, thereby providing a tenable explanation for the detected heart failure association that deserves further exploration in future studies.

Independent experiments support the presence of cardiac muscle enhancer in the identified region. Specifically, ChiP-seq data of p300/CREBBP from an independent cardiomyocyte experiment show a peak at the identified region[16] suggestive of chromatin-accessible active regulatory elements. Since the *ACTN2* gene is known to be induced during cardiomyocyte maturation[17] and our hESC experiments confirm a dynamic regulatory region that switches on during cardiomyocyte differentiation, the absence of evidence of cis-eQTL effects of our putative causal variant with the *ACTN2* gene may reflect a dynamic effect of the enhancer on gene expression during the maturation process or could be the consequence of insufficient power, relevant cell type, or other context-specificity in eQTL studies to-date. Moreover, prior studies support the role of SNPs in this region in cardiac function. Our fine-mapped variant rs535411 is associated with left ventricular end diastolic dimension (beta = 0.022, $t$-test $p$ = 5.07e-05) in a recent large-scale GWAS of echocardiographic traits[18], which corroborates our hypothesis for a role of the locus in left ventricular remodeling. Beyond *ACTN2*, the data from our genetic correlation analysis (Fig. 1b) support a broader role of common variants related to structural musculoskeletal proteins in heart failure by revealing strongly shared heritability between heart failure and multiple musculoskeletal disorders (including osteoarthritis, enthesopathies, intervertebral disk disease) and smooth muscle disorders (esophageal, gastric, and duodenal diseases).

**Regulatory variants of *ABO* predispose to heart failure.** Lastly, the chromosome 9 locus tagged by the SNP rs9411378 is found in an intron of the *ABO* gene, a gene that determines blood type and has been linked to the development of ischemic heart disease[19]. A PheWAS of the sentinel variant across 4155 GWAS from the GWAS Atlas[20] shows its significant effects in hematologic (red blood cell count, white blood cell count, monocyte cell count, hemoglobin concentration) and metabolic traits (lipid disorders, diabetes, activated partial thromboplastin time) (Supplementary Data 5 and Fig. 5a), whereas a similar PheWAS approach on 1448 traits from the UK BioBank reveals its association with venous thromboembolism (Supplementary Fig. 7). Interestingly, conditioning on several traits associated with our sentinel variant for which GWAS summary statistics are available or on known heart failure risk factors does not significantly change the signal of association between the *ABO* locus and heart failure (Table 2 and Supplementary Data 2), suggesting a direct effect of the locus on heart failure independent of its effect on other human disorders. In addition, since *ABO* is a known locus for coronary disease, which in turn is one of the major disorders leading to heart failure, beyond conditioning on ischemic heart disease we also

performed a sensitivity analysis in which we excluded all patients with coronary artery disease (CAD) and tested the association between the locus sentinel SNP and heart failure (log(Odds Ratio) = 0.1027, score test $p$-value = 1.3e-4). Since CAD is only one of the many causes of heart failure, and individuals can have both CAD and heart failure from a different cause, the restricted analysis is conservative and consequently underpowered compared to our discovery GWAS ($N_{cases}$ = 4137 vs. $N_{cases}$ = 10,976), which even at the expected, unrestricted effect size inevitably makes the association non-significant at a genome-wide $p$-value threshold of 5e-8. Nevertheless, the restricted effect size on HF did remain similar to our unrestricted analysis and the association remained nominally significant. Taken together, this sensitivity analysis and the multi-trait conditional analysis suggest the possibility of a role of the *ABO* locus on heart failure independent of its established influence on coronary disease risk. However, definitive proof of this hypothesis will require further study.

The locus did not show any active enhancer or promoter states in cardiac tissues but instead overlapped active enhancer states in primary hematopoietic stem cells and intestinal cells (Supplementary Fig. 8). The sentinel variant was a strong eQTL for *ABO* gene expression in eQTLGen and GTEx whole blood, consistent with our findings of active chromatin state overlap in hematopoietic lineage cells. In addition, the eQTL signal had strong evidence of colocalization with the GWAS signal for the same locus (posterior probability 96%) (Fig. 5b). Notably, the sentinel variant in our GWAS is in LD with the most common variant (rs8176719) associated with O-blood type via a frameshift mutation that is thought to inactivate ABO (LD $r^2$ 0.64 in 1000 Genome Europeans). However, the effects of our lead variants on the expression of *ABO* remain after stratifying by rs8176719 genotype (Fig. 5c), suggesting an additional regulatory role for our variants, which goes beyond tagging the O-blood type variant rs8176719. Similarly, in our GWAS, a strong signal for association remains within the locus after conditioning on rs8176719 (Supplementary Fig. 9). We should note though that rs8176719 is genotyped or accurately imputed only on a small subset of our participants (35,836 individuals), which may limit interpretation of this analysis as definitive evidence of an independent signal. Since our GWAS locus is intronic, we also examined whether it could affect splicing of the *ABO* gene using whole-blood RNA-seq data from GTEx v8. Indeed, although the variant's effect on expression appears stronger than its splicing consequence, we found that the locus is also associated with splicing of *ABO*, promoting a splice variant that skips the exon on which rs8176719 is found, which provides additional evidence for a regulatory role not due to linkage disequilibrium with rs8176719 (Supplementary Table 5 and Supplementary Fig. 10).

Although having non-O blood type via a structural coding variation in the *ABO* gene has been linked to cardiovascular disease and cardiovascular mortality[21], the mechanisms underlying this association are not fully understood. Our finding that regulatory variation of the *ABO* gene's expression is linked to the development of heart failure highlights the importance of *ABO* in cardiovascular disease and opens the door to further studies to decipher the cellular mechanisms involved.

In summary, we performed a large-scale genome-wide association study for heart failure and replicated our findings in a similarly powered cohort. Our results validate the use of this approach to discover regulatory variants associated with heart failure predisposition in response to a variety of cardiac insults, reveal a new putative mechanism for the disease associated with the regulation of a structural cardiac muscle protein during differentiation, underscore the role of the *ABO* gene in cardiovascular disease and highlight broadly shared heritability between heart failure and musculoskeletal disorders.

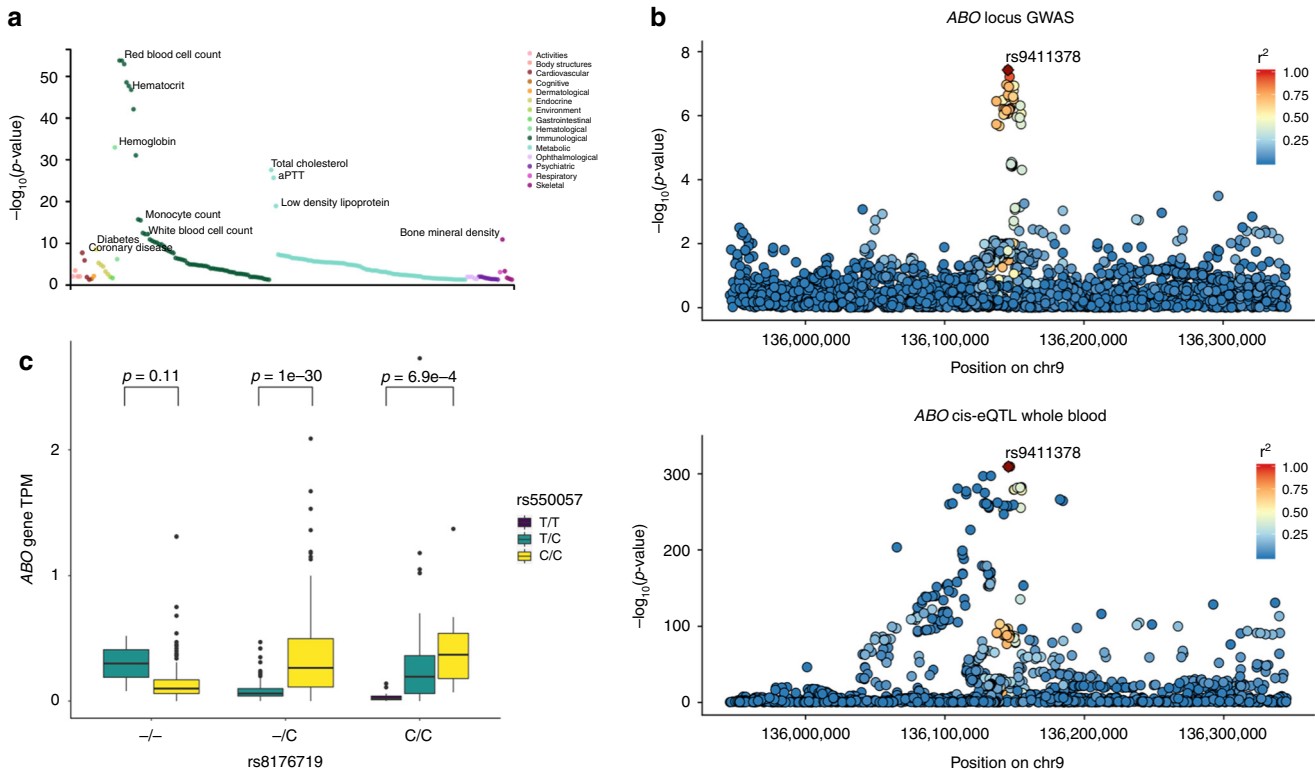

**Fig. 5 Functional characterization of the *ABO* locus. a** Phenome-wide association of the rs9411378 SNP across over 4000 GWAS from the GWAS Atlas. **b** Manhattan plot of the ABO locus overlaying a similar Manhattan plot for the *ABO* gene eQTL in eQTLGen. $r^2$ is the 1000 Genomes phase 3 Europeans LD. We see that the signal colocalizes that is confirmed by coloc (posterior probability of colocalization 96%). **c** Expression transcripts per million for the *ABO* gene in GTEx whole-blood stratified by the genotype of rs8176719 (variant defining blood group O) and rs550057 (variant tagging our sentinel variant rs9411378 (LD $r^2 = 0.92$ in 1000 Genomes Europeans)) ($n = 670$ independent samples). We see that the T allele for rs550057 is associated with non-O blood group, leads to decreased expression of *ABO* in individuals that are not blood group O and denotes increased risk of heart failure in our GWAS (GWAS beta = 0.0961, score test *p*-value = 6.1e-8, $n = 448,549$ independent participants). Boxplot center line represents the median, the bounds represent the interquartile range (IQR) (25–75%) and the whiskers extend from the bounds to the largest value no further than 1.5*IQR from the bound. Data beyond the end of the whiskers are plotted individually. Source data are provided as a Source Data file.

## Methods

**Samples.** We performed genome-wide association studies in five cohorts that study cardiovascular disease (Framingham Heart Study, Cardiovascular Health Study, Atherosclerosis Risk in Communities Study, Multi-Ethnic Study of Atherosclerosis, Women's Health Initiative) and the eMERGE initiative. Genotype and phenotype raw data were downloaded from dbGAP (accession numbers phs000007.v29.p11, phs000287.v6.p1, phs000209.v13.p3, phs000280.v4.p1, phs000200.v11.p3, phs000888.v1.p1). Our work complies with all relevant ethical regulations for work with human participants. All individuals provided informed consent for participation in the individual cohorts. Our GWAS study was approved by the Johns Hopkins School of Medicine IRB (IRB #00163194). For each individual study we performed sample level filtering (excluding samples with assigned and genotype sex discrepancy, extreme deviations from heterozygosity or missingness). We also excluded individuals that were not of European Ancestry and for every group of individuals that were related (identity by descent (IBD) > 0.125) we randomly selected one.

In addition, for each study SNP level filtering was performed to exclude SNPs that had significant deviations from Hardy–Weinberg equilibrium in heart failure controls, minor allele frequency <0.01, missing call rate >0.05 and differential missingness between heart failure cases and controls[22]. For studies that analyzed their populations with different genotyping arrays, we also excluded SNPs that had significant deviation in minor allele frequencies (MAF) between the different arrays. For individuals that were genotyped in more than one genotyping array, we selected the array that had the most extensive genotyping. We proceeded with imputing and analyzing each array separately for every study.

**Imputation.** We imputed each study to the 1000 Genomes phase 3 reference panel using Minimac3[23] after pre-phasing with Eagle[24] on the Michigan Imputation Server. Prior to imputation, we lifted all SNPs to the hg19 human genome build using the UCSC liftOver tool, aligned all SNPs to the positive strand and filtered out SNPs whose minor allele frequencies deviated by >0.2 compared to the reference panel's MAF and SNPs A/T or G/C SNPs with MAF > 0.4 as those are prone to strand alignment errors. After imputation, we excluded all imputed SNPs

with imputation *r* squared (INFO score) <0.7, SNPs with MAF < 0.01 and SNPs with Hardy–Weinberg *p*-value <1e-4. For the eMERGE cohort, imputation was performed independently prior to the start of this study with procedures detailed elsewhere[25] and we subsequently applied the same post-imputation filters.

**Genome-wide association.** For each study, we performed a GWAS for heart failure controlling for age, sex and the first 10 genotype principal components (PCs). Heart failure definitions in the different cohorts are listed in Supplementary Table 6. PCs were calculated based on a set of independent (LD $r^2 < 0.2$) genotyped or high-quality imputed SNPs (INFO score>0.9) in an unrelated population (IBD < 0.08) and the SNP loadings were subsequently used to calculate the eigenvectors for all individuals included in the analysis. In the eMERGE cohort, since the population was collected from multiple different hospitals across the United States, we included an additional multilevel categorical covariate denoting the sample source. All GWAS were performed using a linear mixed model with the saddlepoint approximation (SAIGE)[26] to account for any residual relatedness structure in our analysis and for case–control imbalance, which is inherent in our phenotype of interest. For the UK BioBank cohort we used the summary statistics for all cause Heart Failure (PheCode 428) generated by analyzing the UK BioBank data in the SAIGE paper[26].

**Meta-analysis.** We meta-analyzed the results of all our GWAS using fixed effects inverse-variance meta-analysis via the software METAL[27]. We kept only SNPs that were present in at least three studies and 5000 individuals. The following tools were used for the GWAS: Python, R, Bcftools, PLINK[28], SNPRelate[29], SAIGE[26], METAL[27].

**Replication.** We replicated our findings in an independent cohort of 24,829 Heart failure cases and 1,614,513 controls of European ancestry within the 23andMe research cohort. 23andMe participants provided informed consent and participated in the research online, under a protocol approved by the external AAHRPP-accredited IRB, Ethical and Independent Review Services (E&I Review). Heart

failure in the replication population was self-reported as an answer to the question "Have you ever been diagnosed with or treated for Heart failure?". All three replication variants were imputed with high quality (imputation $r^2 > 0.95$) using an imputation panel that combined the 1000 Genomes Phase 3 panel with the UK10k panel. The variants were analyzed via logistic regression assuming an additive model with covariates for age, sex, the first five genotype PCs, and indicator variables to represent the genotyping platform. The p-values were adjusted for an LD score regression intercept of 1.043.

**Phenome-wide association of heart failure subtypes.** For each of the five cohorts in our study and the eMERGE cohort, we classified heart failure individuals as having ischemic heart failure if they also had a history of diagnosed ischemic heart disease, myocardial infarction, percutaneous coronary intervention or coronary artery bypass graft surgery and non-ischemic heart failure otherwise. We also classified individuals as heart failure with reduced ejection fraction if they had heart failure and at least one echocardiogram showing a left ventricular ejection fraction (LVEF) <50%, and heart failure with preserved ejection fraction otherwise. Individuals that did not have information on myocardial infarction history or echocardiographic information were not included in the respective analyses. We also obtained continuous data of LVEF, left ventricular end diastolic diameter, and interventricular septum diameter from each individual's most recent available echocardiogram. Each of our sentinel variants from the general heart failure GWA meta-analysis were tested for an effect in each of these variables using SAIGE for the categorical variables and linear regression assuming an additive genotype effect for the continuous variables with the same covariates as in our primary GWAS. The results cross-cohort were meta-analyzed using METAL.

**Other phenotype associations.** To evaluate if our lead GWAS variants had associations with other phenotypes we queried the NHGRI-EBI GWAS catalog and also evaluated the GWAS atlas[20], which contains data from 4155 GWAS across 2960 unique traits and the 1488 Electronic Health Record-Derived PheWAS codes from the Michigan Genomics Initiative[26].

**Heritability and genetic correlation.** We used LD score regression[30] with the 1000 Genomes European reference LD to evaluate the liability scale heritability explained by the common variants in our GWAS assuming a population prevalence of 0.018[7]. We subsequently analyzed our GWAS together with summary statistics from GWAS studies from the UK biobank[31] using the genetic correlation method of the LD score regression pipeline to quantify the shared heritability between our phenotype and other traits[32]. For the genetic correlation analysis we selected traits to analyze based on the following procedure:

1. Among all summary statistics analyzed in the SAIGE paper[26], we first excluded the categories Injuries and poisonings (as it is unlikely to have a major heritable component), as well as symptoms and pregnancy complications (as they are too general to have a meaningful interpretation of genetic correlation).
2. We excluded general disease bundles that include the work "other" or "NOS" (e.g., other infectious and parasitic diseases) or are a sign/symptom (e.g., hematuria) or medication (e.g., chemotherapy).
3. We reclassified all infections into the "Infectious Diseases" category and all congenital anomalies to their respective organ system.
4. From every organ system or general disease category we selected the three diseases with the highest number of cases.
5. For every selected disease, we excluded diseases and disorders that are subsets of the same disease or highly related (e.g., Selected disease: hypertension-excluded disease: essential hypertension).
6. We excluded diseases whose z-score of observed heritability calculated via LD score regression was <1.

**Conditional analysis based on summary statistics.** We used the COJO package[33] from the GCTA pipeline to evaluate the residual association signal within our genome-wide significant loci after conditioning on our sentinel variants or other variants of interest using as reference the LD of the eMERGE heart failure dataset.

**Multi-trait conditional and joint analysis.** We used the mtCOJO package[34] from the GCTA pipeline to evaluate the effects of our variants conditioned to other heart failure risk factors (e.g., hypertension, atrial fibrillation, ischemic heart disease) and conditions associated with our sentinel variants in PheWAS studies using the 1000 Genomes Europeans reference LD scores.

**Mendelian randomization analysis.** We used the MR base package[35] to perform Mendelian Randomization analysis in order to evaluate the effect of atrial fibrillation on the development of heart failure using summary statistics from a large-scale GWAS meta-analysis of atrial fibrillation[36]. The polygenic risk score was constructed using independent variants (LD $r^2 < 0.001$) at a genome-wide significance threshold ($p < 5e-8$) using as reference the LD of the 1000 Genomes European samples.

**Variant fine-mapping.** We followed a step-wise approach based on epigenomic annotations and LD structure for our variant fine-mapping efforts. We first used the Roadmap epigenomics ChromHMM 25-state model[13] across all tested cell types and tissues to visualize our significant loci and identify broad patterns of active promoter or enhancer elements across tissues. We subsequently used CAVIAR[12] with the 1000 Genomes European reference LD and the assumption of at most two causal variants per locus to generate a credible set of SNPs for each locus. The CAVIAR analysis for the ACTN2 locus included all SNPs in a radius of 100 kilobases around the locus sentinel SNP. Since that analysis identified a large number of SNPs ($N = 183$), we selected only the 111 SNPs in the set that had a GWAS p-value < 5e-7 for downstream fine-mapping. Then, we intersected the SNPs in that set with active enhancer or promoter elements predicted by Roadmap epigenomics for heart tissues (fetal heart, left ventricle, right ventricle, and right atrium). Finally, we intersected SNPs selected by the previous step with candidate cis-regulatory elements predicted by ENCODE[14]. Bedtools[37] was used for all intersection tests. The WashU Epigenome browser was used for visualization of our loci in the ChromHMM context[38].

**Cardiomyocyte differentiation model.** To further probe the effects of the ACTN2 locus on cardiomyocyte function, we performed ATAC-seq, ChIP-seq of H3K4me3 and H3K27ac, RNA-seq and HiC experiments[39], in an engineered H9 hESC (WiCell Research Institute) modified into H9 hESC MLC2v:H2B-GFP reporter transgenic line, which expresses H2B-GFP in differentiated ventricular cardiomyocytes[40]. This cell line was differentiated into cardiomyocytes using a well-established Wnt-based differentiation protocol[41]. Cardiomyocytes and their intermediate cell populations were collected and analyzed at different differentiation stages (Day 0, 2,5,7,15, and 80) and epigenomics, transcriptomics and three-dimensional chromatin conformation assays were performed on these cells[39]. We queried our fine-mapped variants by intersecting them with ATAC-seq, H3K4me3, and H3K27ac peaks in the cardiomyocyte differentiation model and we subsequently assessed the HiC contacts between the identified peaks and nearby genes. HiC contacts were generated at 5 kb resolution. Expected contacts for each bin are calculated as the genome-wide average of contacts of the same distance, as Hi-C contacts follow a distance-based decay. The observed/expected value for each bin shows the enrichment of HiC contacts relative to the background. We tested the significance of enrichment of observed contacts with respect to the expected contacts using an upper-tail Poisson test with $x$ equals observed contacts and lambda equals expected contacts[42]. The UCSC human genome browser was used to visualize the sequencing peaks[43].

**Expression quantitative trait loci (eQTL).** We assessed the effect of our genome-wide significant variants in gene expression of nearby genes using two databases: (1) Genotype Tissue Expression (GTEx): we obtained whole-genome sequencing and RNA sequencing data from GTEx version 8. We followed the standard pipeline proposed by GTEx v7[44] to normalize gene expression and perform cis-eQTL analyses. In brief, we filtered out genes with <6 reads or <0.1 counts per million (cpm) in >20% of participants per tissue, performed normalization of expression values between samples using TMM[45] and for each gene, we normalized gene expression across samples by an inverse rank-based transform to the standard normal distribution. The effect of a variant in gene expression was analyzed using linear regression as implemented in MatrixEQTL[46] using age, sex, RNA-seq platform, five genotype PCs and 60 probabilistic estimation of expression residuals (PEER) factors[47] as covariates. Using the methods above, we tested the effect of the ACTN2 locus fine-mapped variants to the expression of genes within 1 megabase in left ventricle tissue and the effect of the ABO locus in whole blood. (2) To increase the power of detecting a cis-eQTL association in whole blood, we obtained cis-eQTL summary statistics from eQTLGen, which includes eQTL data from 31,684 samples[48]. We queried our ABO locus sentinel variant in the dataset.

**Colocalization analysis.** For each identified significant eQTL result for our variants, we evaluated whether the eQTL and GWAS signals colocalize using a Bayesian colocalization method as implemented in coloc[49] to estimate the posterior probability of an identical causal variant per locus between eQTL and GWAS. Colocalization Manhattan plots for Supplementary Fig. 1 were generated using LocusZoom[50].

**CRISPR-Cas9 enhancer deletion cardiomyocyte model.** To generate a deletion of the candidate causal region within the ACTN2 locus in human embryonic stem cells (hESCs), we used the CRISPR/Cas9 system. More specifically, we used CHOPCHOP v2[51] to find guide RNAs (gRNAs) that in combination with Cas9 will generate cuts within the ATAC-seq peak detected as causal in our epigenetic hESC-CM experiments. We then cloned both gRNAs and Cas9 in vectors carrying a puromycin resistant cassette and used the NEON electroporation system (ThermoFisher) to effectively transform hESCs (H9 line, Cat# WA09, WiCell). Cells were then plated in flasks coated with Gelltrex (LDEV-Free reduced growth factor basement membrane matrix, Thermofisher) and maintained in Essential 8 medium (Thermofisher). hESCs underwent electroporation using the Thermo Neon Transfection system and transformed hESCs were selected using Puromycin for 48 h. Cells were replated and colonies from single cells were manually picked and expanded. All colonies were subsequently screened using PCR with primers that

bind outside the expected Cas9 cuts. DNA from colonies carrying the deletion generated a 490 bp PCR fragment confirming a ~2200 bp deletion in the target segment (Supplementary Fig. 6B, C). To generate hESC-derived cardiomyocytes, hESCs with enhancer deletion and hESCs from the parent isogenic line were differentiated to cardiomyocytes[52]. For the differentiation protocol, cells were sequentially treated with two small inhibitors, 6 μM of CHIR99021 (Tocris, GSK3b inhibitor) for 48 h followed by 2.5 μM of IWR-1 (Tocris, Wnt signaling antagonist) in RPMI-B27 without insulin medium (Thermofisher). Spontaneous beating was noted at day 7 of differentiation. Cardiomyocytes were further selected using sodium lactate[53]. RNA was isolated from cardiomyocytes at day 15 using Trizol, complementary DNA (cDNA) was generated using the high-capacity cDNA reverse transcription kit (Thermofisher) and quantitative PCR was performed using Sybr select protocol[54]. Gene expression levels were normalized with *GAPDH*. Expression in edited cardiomyocytes and controls was assessed in four replicates and compared with a two-tailed *t*-test. Whole list of primers is provided in Supplementary Table 7.

**Splice-QTL analysis for the *ABO* locus**. We used LeafCutter[55] to perform splice-QTL (sQTL) analysis for rs550057 (which is an LD surrogate for the sentinel SNP rs9411378 of the *ABO* GWAS locus, highly associated with heart failure in the GWAS discovery cohort). We obtained and normalized intron excision ratios from binary sequence alignment/map files for whole-blood tissue in GTEx v8 following the filtering and normalization steps provided with the LeafCutter software. We then used FastQTL[56] to perform nominal sQTL analysis for rs550057 using as covariates five genotype PCs, ten PCs calculated based on the normalized intron excision ratios, along with sex, age, and whole-genome sequencing library construction methodology.

**Reporting summary**. Further information on research design is available in the Nature Research Reporting Summary linked to this article.

## Data availability
Our GWAS summary statistics were made available in a public Zenodo repository (https://zenodo.org/record/3612522#.XiSE_i2ZOgA), and all the genotype data used to generate those summary statistics are available from dbGaP (accession numbers phs000007.v29.p11, phs000287.v6.p1, phs000209.v13.p3, phs000280.v4.p1, phs000200.v11.p3, phs000888.v1.p1) or via a request to the UK BioBank. Our analysis of eQTL data from GTEx and eQTLGen are all available in our Supplementary Tables and the corresponding GTEx v8 sequencing data are available from dbGaP (accession number phs000424.v8.p2) and on the GTEx project portal (https://gtexportal.org/home/). RNA-seq, H3K27ac-seq, and Hi-C data from the cardiomyocyte differentiation experiments have been deposited in the Gene Expression Omnibus under the accession number GSE116862, whereas the sequencing raw reads for ATAC-seq and H3K4me1-seq, as well as all processed epigenetic, RNA-seq, and HiC data in hESC-CMs for our loci of interest were made available at the following Zenodo repository (https://zenodo.org/record/3612522#.XiSE_i2ZOgA). Lastly, the source data underlying Figs. 1b, 3a, c, 4c and Supplementary Figs. 2, 6a, c, 10a are provided as a Source Data file.

## Code availability
The code used for the GWAS analyses is available on the following Github repository: https://github.com/marvani88/HF_GWAS. UCSC genome browser plots were created using the genome browser website (http://genome.ucsc.edu/).

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

## Acknowledgements
We would like to thank the research participants and employees of 23andMe for making this work possible. We would also like to thank François Aguet for providing the processed Leafcutter intron excision ratios for the splicing QTL analysis and Princy Parsana for reviewing the GWAS code. The UK BioBank data was obtained under the UK BioBank resource application 17712. Dr. Arvanitis was supported by NIH T32-HL007227 for this work. Dr. Tampakakis was supported by NIH K08- HL145135-01, AHA 19CDA34660077 and the Johns Hopkins Magic That Matters Fund. Drs. Chatterjeee, Dutta and Battle were supported by NIH R01-HG010480-01.

## Author contributions
M.A. conceptualized and designed the study, performed the analyses, interpreted the results, and wrote the manuscript. E.T. designed and performed the CRISPR hESC-CM experiments and reviewed and revised the manuscript. Y.Z., N. C.C., and B.R. performed the experiments and analyses of the HiC cardiomyocyte differentiation data, interpreted the results, and reviewed and revised the manuscript. W.W., A.A., and the 23andMe Research Team performed the analyses pertaining to GWAS replication, and reviewed and revised the manuscript. D.D. and N.C. performed the sensitivity analysis of the *ABO* locus in individuals without coronary disease, and reviewed and revised the manuscript. S.G. assisted with the CRISPR hESC-CM experiments, and reviewed and revised the manuscript. A.K. contributed to the design of the primary GWAS meta-analysis, and reviewed and revised the manuscript. W.S.P. contributed to the design of the study, the interpretation of the results, and reviewed and revised the manuscript. A.B. conceptualized and designed the study, provided oversight for the analyses, interpreted the results, and reviewed and revised the manuscript.

## Competing interests
W.W., A.A., and the members of the 23andMe Research Team are employees of 23andMe Inc. All other authors declare no competing interests.

## Additional information

## 23andMe Research Team

Michelle Agee[4], Stella Aslibekyan[4], Robert K. Bell[4], Katarzyna Bryc[4], Sarah K. Clark[4], Sarah L. Elson[4], Kipper Fletez-Brant[4], Pierre Fontanillas[4], Nicholas A. Furlotte[4], Pooja M. Gandhi[4], Karl Heilbron[4], Barry Hicks[4], David A. Hinds[4], Karen E. Huber[4], Ethan M. Jewett[4], Yunxuan Jiang[4], Aaron Kleinman[4], Keng-Han Lin[4], Nadia K. Litterman[4], Jennifer C. McCreight[4], Matthew H. McIntyre[4], Kimberly F. McManus[4], Joanna L. Mountain[4], Sahar V. Mozaffari[4], Priyanka Nandakumar[4], Elizabeth S. Noblin[4], Carrie A.M. Northover[4], Jared O'Connell[4], Steven J. Pitts[4], G. David Poznik[4], J. Fah Sathirapongsasuti[4], Anjali J. Shastri[4], Janie F. Shelton[4], Suyash Shringarpure[4], Chao Tian[4], Joyce Y. Tung[4], Robert J. Tunney[4], Vladimir Vacic[4], Xin Wang[4] & Amir S. Zare[4]

