## [Peer Review File · Nature Communications]

Reviewers' comments:

Reviewer #1 (Remarks to the Author):

Arvanitis et al. present the findings from a new GWAS for heart failure. They combine five cohorts to analyze 11k cases / 438k controls and then validated in 23andme data. Despite this large cohort, they only find 3 genome-wide significant loci—which highlights the challenges of CHF for genetic discovery. They then seek to better understand the 3 loci (PITX2, ACTN2, and ABO), and how they mechanistically relate to CHF risk. Larger GWAS for CHF have identified 2 of the 3 loci (PITX2, ABO), along with a few others (<https://www.biorxiv.org/content/10.1101/682013v1>). In both cases the identified loci are also associated with risk factors for CHF. ACTN2 is novel, and could represent a true muscle-related genetic signal that is not dependent on risk factors. Further functional characterization of this signal would be most interesting.

Major Questions:

1. For the discussion of overlap, there are several disorders which make sense based on known risk factors (i.e. CAD, Afib, lipids). It seems that the ABO locus, which is also associated with CAD, is associated with CHF because of this. Does this association go away in cases without CAD?
2. For the ACTA2 story, the authors postulate that the causal SNP (rs535411) does not have an eQTL for ACTN2 because it plays an important role in more embryologic cardiomyocytes, like their stem cell derived cells. This is a reasonable and very interesting hypothesis, but they stop one step short of proving it. Do isogenic stem cell-derived CMs that have been edited at the causal SNP display differential expression of ACTN2?
3. Could the signal be a synthetic association at ACTN2 locus? Given that there are case reports of coding mutations associated with dilated and hypertrophic cardiomyopathy?
4. Afib and CHF: causation versus reverse causation? Can the later be specifically excluded? It is not clear to me from the analysis that Afib causes CHF instead of CHF causing Afib.
5. Figure 1B does not include Afib, and instead has cardiac dysrhythmias, which is a much broader phenotype. Text on line 89 says that Afib showed genetic correlation. Was it specifically Afib, or all dysrhythmias?

Rajat Gupta (Reviewer)

Reviewer #2 (Remarks to the Author):

Arvanitis et al. present a soundly designed and well-conducted GWAS meta-analysis of heart failure using five large clinical and population genetics cohorts. They identify three loci, two of which are novel, and replicated these findings in a cohort of 23andMe participants. Using this pooled cohort,

they also evaluated shared heritability of HF with other complex traits. They go on to evaluate each identified locus for its putative causality: For PITX2 (previously identified in a HF cohort) they use both multi-trait conditional and joint analyses to show that its known causal relationship with atrial fibrillation likely drives its association with heart failure. They use genetic data from a pooled cohort to demonstrate that AF can cause HF using Mendelian Randomization. For ACTN2, they do a good job showing this locus's effect on multiple HF causes and ventricular remodeling using PheWAS. Their initial eQTL analysis did not show any association of the locus with specific gene expression, but they do a very nice job demonstrating that this locus is active during hESC cardiomyocyte differentiation and that it binds the ACTN2 promoter. For the locus in the ABO intron, they perform PheWAS to show its effect in hematologic and metabolic traits, and show that conditioning on these traits does not change the association they originally identified with HF. Their sentinel variant is a strong eQTL for ABO, but is also in LD with the most common variant associated with O-blood type due to a frameshift variant in ABO (though adjusting for this variant did not completely obfuscate its effect on ABO expression.)

Overall, this study is a methodologically sound report of the largest meta-analysis of GWAS data for heart failure to date and finds one very interesting and regulatory locus associated with ACTN2 as well as a previously identified locus associated with PITX2 and another novel locus associated with the ABO gene. A few questions remain to be addressed:

- What were the inclusion criteria for HF diagnosis in each of the five cohorts?
- 23andMe is a popular service and it's not clear whether there is any potential for overlap between this replication cohort and the discovery cohort. Assuming the 23andMe cohort is from consented participants), what is the likelihood that there is overlap? Would also be helpful to provide as much demographic information on the 23andMe cohort as possible in comparison to the discovery cohort. Should also probably state something about the 23and Me consent process/IRB approval in methods. If there is a publication that better describes the 23andMe cohort, it would also be fine to point the reader to that instead.
- The authors state clearly that ACTN2 expression is regulated in hESC differentiation [14] – was it turned on around day 8 in those studies as well? Would be nice to see a correlation between your time course findings and the actual start of ACTN2 expression.
- What evidence exists that variation in ACTN2 expression can modulate relevant cellular phenotypes? If none exists, please provide in vitro evidence that modulation of ACTN2 expression changes a heart failure-relevant cardiomyocyte phenotype (e.g. size, NPPB expression) in hESC-derived CMs.
- Lines 123-125 “Integration with eQTL data does not reveal any compelling evidence of colocalization between the GWAS signal and altered expression of nearby genes in adult blood or post-mortem adult heart tissues (Supplemental Table 4).” As the authors know, post-mortem tissue from GTEX is fraught with issues of accurate gene expression. A few studies have been done now on either freshly preserved tissues at the time of heart transplant/donor heart explant or from biopsies of patients with DCM. Please report whether these studies have reported any association of this

locus with genes of interest: PMID: 29138229 (eQTLs available at https://ccb-web.cs.uni-saarland.de/cms/results_csvs/), and PMID 31235787 (eQTLs available at: <https://zenodo.org/record/2617028#.XWMJHZNKiQU>).

- Is there any reason to believe that the ABO intronic locus affects ABO splicing? This could be tested fairly easily and would provide additional convincing evidence that the LD with the known frameshift variant is not the major cause for the association as well as point to a potential mechanism, especially if the previously reported frameshift variant is in an exon that could be affected by differential splicing. If this type of in vitro work is out of scope for the authors, it would be interesting to know if the variant lies in any predicted splice sites.

Minor Comments:

- Please make fonts bigger on all figures including the supplementary figures.
- For review purposes, it helps to label the figures with figure numbers for ease of navigation.
- While ACTN2 is inside the sarcolemma, it's main role is to form connections between actin filaments in the sarcomere by binding to actin with one terminus and then dimerizing at the other. It seems a bit odd to refer to it as a sarcolemmal protein.
- Figure 1B – I don't find the significant GI-related diseases or musculoskeletal diseases on the plot? How did you group these? For some reason, I don't have access to Supplemental Table 2, so, perhaps this is where this data is featured. But since you circle back to the musculoskeletal hypothesis, might consider bringing this into the main figure.
- Figure 2 – would be easier to label samples in a way that the reader can easily understand them – e.g. re-label ATAC_D*** to "ATAC-seq Day ***".

Reviewer #3 (Remarks to the Author):

In the manuscript entitled "Genome-wide association and multi-OMIC analyses reveal new mechanisms for Heart failure" the authors performed genome-wide association meta studies of multiple cohorts with heart diseases. The authors replicate the findings in a population cohort. Two loci have been further characterized in terms of their potential functional relevance and possible mechanistic explanations have been provided.

There are conceptual issues with this study that are partly due to the nature of meta-analysis, but also due to the design of their general strategy and functional work-up.

- The use of all cardiovascular traits as proxies of heart failure will likely result in higher rates of false negatives and might suggest to the reader that previous findings from original papers cannot be replicated. However, this is due to the assumptions of the authors that every cardiovascular trait is involved in heart failure (either as risk factor, co-morbidity or co-causal factor as shown for afib). As an example of the problematic strategy the reviewer refers to Figure 1B, where you find significant associations between "abdominal hernia" and "ischemic heart failure" or "arrhythmia". This makes little sense and will not propel the research on heart failure.
- The p-values for screening and replication are quite modest assuming the cohort sizes and also

underline the conceptual problem of this approach.

- The “multi-omics” analysis is introducing a positive selection bias in its current form and it seems like picking raisins. Both investigations on ABO and ACTN2 are underlined by highly selective strategies using diverse datasets and methods to underline positive associations on genetic, epigenetic and partially in-vitro studies. This part has to be redone completely and performed on the whole genome-wide SNP data, all in the same manner and corrected for multiple testing in this setting.
- The title should be changed completely. The main features are “meta-analysis”, “cardiovascular traits” and “associations”. Everything else is clearly an overstatement.
- More evidence at the protein level and the “real” functional level must be provided for the ACTN2 and ABO loci to strengthen their role in heart failure as suggested in the title.
- In addition, the supplemental material should contain the scripts used for the GWAS, preferably a Python Jupyter notebook for more transparency.

General Response to all Reviewers

We thank the reviewers for their response to our manuscript. We appreciate their helpful and thoughtful comments and have incorporated their feedback into our revised paper. Based on the recommendations, we made several revisions to our manuscript that we believe improve the quality and impact. The most significant changes include:

1. We expanded the functional characterization and follow up of the ACTN2 locus, specifically showing a temporal correlation between the appearance of our identified ATAC peak and the expression of ACTN2 in cardiomyocytes.
2. We provide experimental validation that our predicted enhancer element at the ACTN2 locus does indeed impact ACTN2 expression during cardiomyocyte differentiation. We performed a CRISPR knock-out of the putative enhancer in human embryonic stem cells and demonstrated that ACTN2 expression is reduced by approximately half on day 15 of cardiomyocyte differentiation compared to isogenic controls without the deletion.
3. We expanded characterization of the ABO locus by showing that the signal for an effect on heart failure persists even when we restrict analysis to the subset of our GWAS participants that had no evidence of coronary disease, supporting that this locus acts to affect heart failure independent of coronary disease. Additionally, we showed that our locus variants may influence the splicing of ABO in whole blood.
4. A final minor note: in the course of revising our analysis, we noticed a minor discrepancy in the LD matrix generated by plink used for our CAVIAR analysis. Correcting this led to some additional variants being included in the credible set of SNPs for the ACTN2 locus, but it did not ultimately affect any of our downstream results or conclusions.

We thank the Reviewers for their useful feedback and the opportunity to improve our paper. An itemized response follows:

Reviewer #1

Overall Comment: Arvanitis et al. present the findings from a new GWAS for heart failure. They combine five cohorts to analyze 11k cases / 438k controls and then validated in 23andme data.

Despite this large cohort, they only find 3 genome-wide significant loci—which highlights the challenges of CHF for genetic discovery. They then seek to better understand the 3 loci (PITX2, ACTN2, and ABO), and how they mechanistically relate to CHF risk. Larger GWAS for CHF have identified 2 of the 3 loci (PITX2, ABO), along with a few others (<https://www.biorxiv.org/content/10.1101/682013v1>). In both cases the identified loci are also associated with risk factors for CHF. ACTN2 is novel, and could represent a true muscle-related genetic signal that is not dependent on risk factors. Further functional characterization of this signal would be most interesting.

Response: We would like to thank the reviewer for their favorable view of our manuscript and for their constructive comments and suggestions. As detailed in our point-by-point responses, we performed further functional characterization of the ACTN2 locus signal according to their suggestions.

Comment #1: For the discussion of overlap, there are several disorders which make sense based on known risk factors (i.e. CAD, Afib, lipids). It seems that the ABO locus, which is also associated with CAD, is associated with CHF because of this. Does this association go away in cases without CAD?

Response: This is a great point. Since ABO is a known locus for coronary disease one could naturally assume that its association with heart failure is mediated through its association with CAD. Although that would be a reasonable explanation, it does not seem to be fully supported by our data and analyses. Specifically, when we condition on ischemic heart disease the association with heart failure remains highly significant (contrast that with the association between the PITX2 locus and HF, which is completely eliminated when we condition on Afib). In addition, as per your suggestion, we performed a sensitivity analysis in which we excluded all patients with CAD and performed a GWAS meta-analysis for heart failure on the subset of patients without CAD in our discovery cohort. The effect size remains nearly the same ($\beta=0.1027$, compared to the original $\beta=0.1142$) in this restricted analysis. The reduced sample size ($N_{\text{cases}}=4,137$ vs $N_{\text{cases}}=10,976$) limits power in this analysis, naturally diminishing the p-value somewhat as expected, but the consistent effect size supports an independent effect of the locus on HF. Lastly, we note that if the association between ABO and HF was indeed mediated fully via CAD, one would naturally expect loci with a stronger signal for CAD association (such as 9p21, PHACTR1, PSRC1/SORT1, LPA/PLG to name just a few) to also be more strongly associated with the heart failure phenotype than ABO.

We revised our manuscript to reflect this discussion and additional analysis as follows:

Page 8 Paragraph 1: *“Interestingly, conditioning on several traits associated with our sentinel variant for which GWAS summary statistics are available or on known heart failure risk factors does not significantly change the signal of association between the ABO locus and heart failure (Table 2, Supplemental Table 4), suggesting a direct effect of the locus on heart failure independent of its effect on other human disorders. In addition, since ABO is a known locus for coronary disease, which in turn is one of the major disorders leading to heart failure, beyond conditioning on ischemic heart disease we also performed a sensitivity analysis in which we excluded all patients with coronary artery disease (CAD) and tested the association between the locus sentinel SNP and heart failure ($\log(\text{Odds Ratio})=0.1027$, $p\text{-value}= 1.3e-4$). The restricted analysis is underpowered compared to our discovery GWAS ($N_{\text{cases}}=4,137$ vs $N_{\text{cases}}=10,976$), but the association signal persists with a similar effect size to the unrestricted analysis, providing further support for an effect of ABO on heart failure independent of its effect on CAD.”*

Comment #2: For the ACTA2 story, the authors postulate that the causal SNP (rs535411) does not have an eQTL for ACTN2 because it plays an important role in more embryologic cardiomyocytes, like their stem cell derived cells. This is a reasonable and very interesting hypothesis, but they stop one step short of proving it. Do isogenic stem cell-derived CMs that have been edited at the causal SNP display differential expression of ACTN2?

Response: We agree with the reviewer that it is important to show that our identified putative enhancer element affects ACTN2 expression during differentiation. Therefore, we developed engineered hESC cardiomyocytes with a deletion of the enhancer that was detected as likely causal in our multi-omic analyses using CRISPR-Cas9 editing. Using this model we validate that ACTN2 expression is indeed significantly reduced by half in the engineered cardiomyocytes compared to their isogenic controls on day 15 of differentiation (see Figure 3C in our revised manuscript). We subsequently tested all genes within a megabase of the sentinel ACTN2 locus SNP in this engineered cardiomyocyte model and we show that ACTN2 appears to be the only proximal gene affected by the deletion, thereby confirming that this region is an enhancer specific for ACTN2 in differentiating cardiomyocytes and providing a plausible explanation for its effects in heart failure. We note that we chose to test the full enhancer deletion rather than a single SNP for three key reasons: 1) we were primarily seeking to validate that the computationally predicted regulatory element within our identified GWAS locus is indeed an enhancer that influences ACTN2 expression during cardiomyocyte differentiation, 2) a full deletion of the enhancer is more likely to result in a detectable expression change from a tractable number of replicates compared to a single small-effect SNP, which may require dozens or hundreds of replicates to achieve significance, and 3) CRISPR edits at single nucleotide loci are more time-consuming to produce and validate, and in the time frame of this manuscript, the

full enhancer deletion was more plausible. We made substantial revisions in our methods and main text and added a plot in Figure 3C and D to incorporate the data summarized above. The main changes are as follows:

Page 7 Paragraph 2: *“We subsequently ventured to experimentally validate the effect of the putative enhancer element at the 1q43 locus on ACTN2 gene expression in cardiomyocytes. For that purpose, we generated engineered hESCs with a CRISPR-Cas9 induced deletion in the ~2200bp region that delimits the enhancer element identified in our hESC-CM epigenomic data analyses. We then differentiated these edited hESCs into cardiomyocytes and on day 15 of differentiation, we compared the expression of ACTN2 to that of isogenic hESC-CMs without the deletion. ACTN2 expression was reduced on average by half in the edited hESC-CMs compared to controls (Figure 3C). We then assessed expression of other nearby genes, and none appeared to be affected by the deletion (Supplemental Figure 6C). These experiments support the epigenetic predictions of a cardiac enhancer element in that region and validate the Hi-C data that suggest binding of that enhancer element to the ACTN2 gene promoter. More importantly, these results provide a mechanistic hypothesis of the GWAS association between the ACTN2 locus and heart failure.”*

AND

Page 15 Paragraph 3: *“To generate a deletion of the candidate causal region within the ACTN2 locus in human embryonic stem cells (hESCs), we used the CRISPR/Cas9 system. More specifically, we used CHOPCHOP v2 [47] to find guide RNAs (gRNAs) that in combination with Cas9 will generate cuts within the ATAC-seq peak detected as causal in our epigenetic hESC-CM experiments. We then cloned both gRNAs and Cas9 in vectors carrying a puromycin resistant cassette and used the NEON electroporation system (ThermoFisher) to effectively transform hESCs (H9 line). Cells were then plated and maintained as previously described and transformed hESCs were selected using Puromycin for 48 hours. Cells were replated and colonies from single cells were manually picked and expanded. All colonies were subsequently screened using PCR with primers that bind outside the expected Cas9 cuts. DNA from colonies carrying the deletion generated a 490bp PCR fragment confirming a ~2200bp deletion in the target segment (Supplemental Figure 6 B and C). To generate hESC-derived cardiomyocytes, hESCs with enhancer deletion and hESCs from the parent isogenic line were differentiated to cardiomyocytes as previously described [48]. Spontaneous beating was noted at day 7 of differentiation. Cardiomyocytes were further selected using sodium lactate [49]. RNA was isolated from cardiomyocytes at day 15 using Trizol and quantitative PCR was performed as previously described [50]. Gene expression levels were normalized with GAPDH. Expression in*

edited cardiomyocytes and controls was assessed in four replicates and compared with a two-tailed t-test. Whole list of primers is provided in Supplemental Table 12.”

Comment #3: Could the signal be a synthetic association at ACTN2 locus? Given that there are case reports of coding mutations associated with dilated and hypertrophic cardiomyopathy?

Response: Thank you for this comment. Indeed, there are multiple case reports in the literature for coding mutations in the ACTN2 gene that lead to both dilated and hypertrophic cardiomyopathy and consequently heart failure which further support a central role for this gene in the development of the disease. Although these coding variants are rare, it is not implausible that they could provide an additional signal for association through LD with nearby variants. This is not a possibility that we can completely exclude since we are unable to directly test rare variants in our study because of the lack of whole genome sequencing data in the majority of our participants. However, there is some evidence that the effect of these rare variants even if present does not account for the majority of our detected association signal within the ACTN2 locus. First, in our credible set detection analysis via CAVIAR, we note that with >95% probability, the association signal comes from variants within a DNA region before the start of the ACTN2 gene. In addition, conditioning on our sentinel variant, largely eliminates the association signal in the locus suggesting that variants in incomplete LD with the sentinel SNP are not the major contributors of the noted association. That said, we believe this is an important point to clarify and have therefore changed the text to incorporate that discussion as follows:

Page 6 Paragraph 4: *“Although rare variants within the ACTN2 gene are known to be associated with cardiomyopathies, the credible set analysis does not support the coding region as being the primary driver of the GWAS signal. Moreover, conditioning on the sentinel variant eliminates the signal for association of the locus with heart failure, suggesting that the association is driven primarily by a single causal variant in high LD with the sentinel SNP (Supplemental Figure 5). We should note however that we cannot exclude the possibility that the association signal could be caused or increased by other rare variants within our identified cardiac muscle enhancer region that contains rs535411.”*

Comment #4: Afib and CHF: causation versus reverse causation? Can the later be specifically excluded? It is not clear to me from the analysis that Afib causes CHF instead of CHF causing Afib.

Response: This is a very important comment that deserves further clarification so we would like to thank the reviewer for raising this point. Current clinical understanding supports a bi-causal

relationship between atrial fibrillation and heart failure, via the potential development of tachycardia mediated cardiomyopathy from longstanding rapid atrial fibrillation, and conversely, the onset of atrial fibrillation as a consequence of longstanding elevation of filling pressures and atrial stretching associated with heart failure (PMID 19433768). As noted by the reviewer, our Mendelian Randomization analysis of the effect of atrial fibrillation on heart failure was performed to evaluate the assumption that atrial fibrillation may be causal to the development of heart failure. We were not able to test the potential opposite causal direction, primarily because we are limited by lack of a robust instrument, due to the small number of genome-wide significant variants associated with heart failure in this or previous studies. It is important to note that we performed our Mendelian Randomization using a multiple variant approach, so the result does not hinge on a single variant, which would be more susceptible to ambiguity in directionality. Specifically, we collected all (N=110) genome wide significant loci for atrial fibrillation, and constructed a polygenic risk score for Afib based on these. We then assessed the effect of the Afib polygenic score on heart failure. Although horizontal pleiotropy cannot be completely excluded in any Mendelian Randomization study, we should note that sensitivity analyses using the weighted median, mode and MR Egger estimates for Mendelian randomization that are known to largely account for pleiotropic effects continue to show a significant MR causal estimate for atrial fibrillation on heart failure (see PMID 30002074 for a detailed discussion of horizontal pleiotropy and the methods above). Notably, weighted median requires only 50% of the variants to be valid instruments for the MR estimate to be consistent (PMID 27061298).

MR approaches do not themselves rule out reverse causation, even with a multivariate approach. However, other lines of evidence suggest reverse causation is unlikely to explain this MR result. First, in our HF GWAS and previous studies of HF, SNP heritability estimates were quite low in comparison to the strong heritability observed in Afib GWAS. Second, effect sizes for HF are small, and in fact >95% (105/110) of the tested variants in our Mendelian Randomization have a higher effect size estimate for Afib than for heart failure, suggesting an effect of these variants in Afib that is not dependent on heart failure. Thus, while reverse causation is still technically possible, it seems unlikely that the majority of tested atrial fibrillation GWAS variants are actually directly associated with heart failure and only lead to atrial fibrillation via reverse causation. In summary, our MR results combined with the observed effect sizes in Afib and HF GWAS support the causal effect from Afib on heart failure based on a multi-variate MR, but do not rule out (or assess) a separate causal effect from heart failure on Afib, because we lack power to detect this.

We revised the text to highlight this issue as follows:

Page 5 Paragraph 2: “Mendelian randomization (MR) analysis using 110 independent ($LD\ r^2 < 0.001$) genome-wide significant atrial fibrillation-associated variants, provides further evidence for a directional effect of atrial fibrillation on heart failure development (weighted mode MR effect size 0.21 (Odds Ratio 1.23), $p < 0.0001$). Sensitivity analysis using the MR Egger and weighted median approach to account for potential pleiotropy and/or invalid instruments confounding our MR estimates is statistically significant and supports our hypothesis for a causal effect of atrial fibrillation on heart failure (Supplemental Figure 2). While MR methods alone cannot rule out reverse causation, 105/110 of the variants used here have a larger effect size for atrial fibrillation than heart failure, and atrial fibrillation displays greater SNP heritability, supporting that the MR result indicates an effect on Heart Failure that is mediated through atrial fibrillation rather than the reverse.”

Comment #5: Figure 1B does not include Afib, and instead has cardiac dysrhythmias, which is a much broader phenotype. Text on line 89 says that Afib showed genetic correlation. Was it specifically Afib, or all dysrhythmias?

Response: Thank you for this comment. This is a mistake on our part and we apologize for the confusion. Beyond genetic correlation of heart failure with the three most prevalent traits from each organ system category, we had also run genetic correlations with other cardiovascular disorders that we inadvertently left out of Supplemental Table 2. Those tests included atrial fibrillation which does show significant genetic correlation with heart failure ($rg\ 0.4949$, $p\text{-value}\ 1.663e-18$) as do cardiac dysrhythmias in general. We revised Supplemental Table 2 to reflect that.

Reviewer #2

Overall Comment: Arvanitis et al. present a soundly designed and well-conducted GWAS meta-analysis of heart failure using five large clinical and population genetics cohorts. They identify three loci, two of which are novel, and replicated these findings in a cohort of 23andMe participants. Using this pooled cohort, they also evaluated shared heritability of HF with other complex traits. They go on to evaluate each identified locus for its putative causality: For PITX2 (previously identified in a HF cohort) they use both multi-trait conditional and joint analyses to show that its known causal relationship with atrial fibrillation likely drives its association with heart failure. They use genetic data from a pooled cohort to demonstrate that AF can cause HF using Mendelian Randomization. For ACTN2, they do a good job showing this locus’s effect on multiple HF causes and ventricular remodeling using PheWAS. Their initial eQTL analysis did not show any association of the locus with specific gene expression, but they do a very nice job demonstrating that this locus is active during hESC cardiomyocyte differentiation and that it

binds the ACTN2 promoter. For the locus in the ABO intron, they perform PheWAS to show its effect in hematologic and metabolic traits, and show that conditioning on these traits does not change the association they originally identified with HF. Their sentinel variant is a strong eQTL for ABO, but is also in LD with the most common variant associated with O-blood type due to a frameshift variant in ABO (though adjusting for this variant did not completely obfuscate its effect on ABO expression.)

Overall, this study is a methodologically sound report of the largest meta-analysis of GWAS data for heart failure to date and finds one very interesting and regulatory locus associated with ACTN2 as well as a previously identified locus associated with PITX2 and another novel locus associated with the ABO gene. A few questions remain to be addressed.

Response: We would like to thank the reviewer for their favorable review of our study design and findings. We appreciate their comments as they allowed us to improve our manuscript and have incorporated their suggestions.

Comment #1: What were the inclusion criteria for HF diagnosis in each of the five cohorts?

Response: Thank you for identifying a need to clarify that for the reader. We added a table in our supplemental data (Supplemental Table 11) to reflect the inclusion criteria for HF in each cohort as per your suggestion. In addition, we list the definitions in the table below for your convenience.

Cohort	Heart Failure Definition
ARIC	Hospitalization with a heart failure diagnosis according to ICD codes in any position or a death certificate with death from heart failure in any position
CHS	The participant must have both a congestive heart failure diagnosis by a physician and be under treatment with medications for congestive heart failure
Framingham	A definite diagnosis of congestive heart failure requires that a minimum of two major or one major and two minor criteria be present concurrently. The presence of other conditions capable of producing the symptoms and signs are considered in evaluating the findings.
MESA	Heart failure presence adjudicated by MESA investigators based on presence of symptoms and imaging findings attributable to heart failure along with a diagnosis of heart failure by a physician and medical treatment for heart failure.
WHI	The participant must have both a congestive heart failure diagnosis by a physician and be under treatment with medications for congestive heart failure
eMERGE	Presence of ICD9 codes for heart failure and positive mention of heart failure in the participant's

	problem list based on either natural language processing or a structured problem list.
UK Biobank	Hospitalization with a heart failure diagnosis according to ICD codes in any position

Comment #2: 23andMe is a popular service and it’s not clear whether there is any potential for overlap between this replication cohort and the discovery cohort. Assuming the 23andMe cohort is from consented participants), what is the likelihood that there is overlap? Would also be helpful to provide as much demographic information on the 23andMe cohort as possible in comparison to the discovery cohort. Should also probably state something about the 23and Me consent process/IRB approval in methods. If there is a publication that better describes the 23andMe cohort, it would also be fine to point the reader to that instead.

Response: Thank you for this comment. As per your suggestion, we added some relevant demographic information in the comparison between the Discovery and Replication cohorts in a table (Supplemental Table 2). Since the replication cohort is > 95% US-based while a significant proportion of the discovery cohort (~90%) is UK-based and because recruitment for all the different cohorts was independent among a large pool of European ancestry individuals, the potential for significant overlap is minimal. That said, as it is not feasible for us to identify specific individuals who may be participants in both cohorts we cannot completely eliminate the possibility of a small overlap. We included a sentence in our manuscript to acknowledge this. In addition, we revised our methods section to include some information about IRB approval for the 23andMe cohort as per your recommendation. The revised text reads:

Page 4 Paragraph 2: *“Demographic information comparing the Discovery and Replication cohorts is available in Supplemental Table 2.”*

AND

Page 11 Paragraph 3: *“23andMe participants provided informed consent and participated in the research online, under a protocol approved by the external AAHRPP-accredited IRB, Ethical and Independent Review Services (E&I Review).”*

AND

Supplemental Table 2: *“Although it is unlikely based on the relative demographics of cohort participants, we should note that we cannot exclude the possibility that a small percentage of our Discovery cohort participants are also part of the 23andMe cohort”*

Comment #4: The authors state clearly that ACTN2 expression is regulated in hESC differentiation [14] – was it turned on around day 8 in those studies as well? Would be nice to see a correlation between your time course findings and the actual start of ACTN2 expression.

Response: This is an interesting question. In previous studies of stem cell- cardiomyocyte differentiation, ACTN2 expression is induced around day 6 (see for example PMID 29337667 Figure 1). We agree that showing this time correlation strengthens the argument for an effect of our identified ATAC-seq peak and ACTN2 expression and we therefore expanded our Figure 2 to show RNA-seq data from our hESC-CM experiments that reveal a clear correlation between the timing of the expression of the ACTN2 gene and the appearance of the ATAC-seq peak (Figure 2C), where they appear to arise together around days 5-7. The exact relative timing cannot be fully determined given a resolution of data collection at 6 differentiation time points.

Comment #5: What evidence exists that variation in ACTN2 expression can modulate relevant cellular phenotypes? If none exists, please provide in vitro evidence that modulation of ACTN2 expression changes a heart failure-relevant cardiomyocyte phenotype (e.g. size, NPPB expression) in hESC-derived CMs.

Response: We agree with the reviewer that such a discussion would solidify our findings. It is well established that missense mutations of the ACTN2 gene can lead to both dilated and hypertrophic cardiomyopathies in humans (PMID 23281406 and 20022194). Additionally, we agree that it is important to showing that a decrease in ACTN2 expression (not only protein coding changes) can lead to cellular and organismal phenotypes. To that end, we included a brief discussion of a recent paper in the literature (PMID 22253474) which shows that ACTN2 gene knockdown (reduced expression by 85%) leads to multiple phenotypic changes in zebrafish including decreased number and size of cardiac sarcomeres, enlarged hearts with thin walls and a decreased heart rate. Our revised text reads:

Page 7 Paragraph 2: *“Indeed, previous studies have established that reduction of ACTN2 mRNA levels via a siRNA leads to defects in the number and size of cardiac sarcomeres along with a phenotype of dilated heart with thin walls and a decreased heart rate in zebrafish [14]. It is therefore plausible that smaller reductions of ACTN2 expression as those caused by variants within our identified enhancer could generate subtler cardiac sarcomeric defects in humans that*

become apparent later in life in individuals with an additional genetic or environmental insult to the heart muscle, thereby providing a tenable explanation for the detected heart failure association that deserves further exploration in future studies.”

Comment #6: Lines 123-125 “Integration with eQTL data does not reveal any compelling evidence of colocalization between the GWAS signal and altered expression of nearby genes in adult blood or post-mortem adult heart tissues (Supplemental Table 4).” As the authors know, post-mortem tissue from GTEX is fraught with issues of accurate gene expression. A few studies have been done now on either freshly preserved tissues at the time of heart transplant/donor heart explant or from biopsies of patients with DCM. Please report whether these studies have reported any association of this locus with genes of interest: PMID: 29138229 (eQTLs available at https://ccb-web.cs.uni-saarland.de/cms/results_csvs/), and PMID 31235787 (eQTLs available at:

<https://zenodo.org/record/2617028#.XWMJHZNKiQU>).

Response: We would like to thank the reviewer for pointing us to these new eQTL datasets, which we agree could show improved signal in heart tissue. We tested the effect of our putative causal variant in the ACTN2 locus on expression of nearby genes in the eQTL data from PMID 31235787, however we were unable to find clear evidence for association with gene expression at a Bonferroni adjusted level of 0.05 in either controls or cases with DCM. Unfortunately, our putative causal variant was not tested in the other reference provided by the reviewer (PMID 29138229). It is possible that the lack of association reflects lack of power to show an effect (N=136 non-failing donor hearts), in fact the nominal association p-value between rs535411 and ACTN2 expression in controls in PMID 31235787 is 0.0069 (but is not significant level when adjusted for multiple testing). Larger studies may eventually reveal an effect in adult tissue. Alternatively, this could be an effect that is present mostly during differentiation and therefore is difficult to detect in the adult heart. We revised our text to discuss these possibilities as follows:

Page 6, Paragraph 1: *“In addition, no association with the expression of nearby genes is detected in eQTL studies performed with freshly preserved heart tissue at the time of heart transplant/donor heart explant [10].”*

AND

Page 7, Paragraph 3: *“Since the ACTN2 gene is known to be induced during cardiomyocyte maturation [16] and our hESC experiments confirm a dynamic regulatory region that switches on during cardiomyocyte differentiation, the absence of evidence of cis-eQTL effects of our*

putative causal variant with the ACTN2 gene may reflect a dynamic effect of the enhancer on gene expression during the maturation process or could be the consequence of insufficient power, relevant cell type, or other context-specificity in eQTL studies to-date.”

Comment #7: Is there any reason to believe that the ABO intronic locus affects ABO splicing? This could be tested fairly easily and would provide additional convincing evidence that the LD with the known frameshift variant is not the major cause for the association as well as point to a potential mechanism, especially if the previously reported frameshift variant is in an exon that could be affected by differential splicing. If this type of in vitro work is out of scope for the authors, it would be interesting to know if the variant lies in any predicted splice sites.

Response: This is a very intriguing hypothesis by the reviewer and we would like to thank them for the suggestion. We were able to computationally evaluate the effect of our intronic ABO locus SNPs on alternative splicing of ABO using whole blood RNA sequencing data from GTEx v8, with splice junction usage evaluated using Leafcutter software. Indeed, we discovered that our intronic variants may affect splicing of the ABO gene, promoting a splicing transcript that skips the exon on which the frameshift variant rs8176719 is located. We agree with the reviewer that this provides additional supportive evidence that the LD with rs8176719 is not the major cause for the associations we see for this pleiotropic locus. We revised our text to include the additional findings:

Page 9, Paragraph 1: *“Since our GWAS locus is intronic, we also examined whether it could affect splicing of the ABO gene using Whole Blood RNA-seq data from GTEx v8. Indeed, although the variant’s effect on expression appears stronger than its splicing consequence, we found that the locus is also associated with splicing of ABO, promoting a splice variant that skips the exon on which rs8176719 is found, which provides additional evidence for a regulatory role not due to linkage disequilibrium with rs8176719 (Supplemental Table 10, Supplemental Figure 10).”*

AND

Page 16, Paragraph 2: *“We used LeafCutter [51] to perform splice-QTL (sQTL) analysis for rs550057 (which is an LD surrogate for the sentinel SNP rs9411378 of the ABO GWAS locus, highly associated with heart failure in the GWAS discovery cohort). We obtained and normalized intron excision ratios from binary sequence alignment/map files for Whole Blood tissue in GTEx v8 following the filtering and normalization steps provided with the LeafCutter software. We then used FastQTL [52] to perform nominal sQTL analysis for rs550057 using as covariates 5*

genotype PCs, 10 PCs calculated based on the normalized intron excision ratios, along with sex, age and whole genome sequencing library construction methodology.”

Minor Comments:

Comment #8: Please make fonts bigger on all figures including the supplementary figures.

Response: We increased the font in all main and most supplemental figures to make them more readable as per your suggestion.

Comment #9: For review purposes, it helps to label the figures with figure numbers for ease of navigation.

Response: We were not sure how to add numbers in the main figures only for the purposes of review so we opted against that in order to avoid having the numbers printed in the final manuscript. However, we labeled all supplemental figures with a number as per your suggestion.

Comment #10: While ACTN2 is inside the sarcolemma, it's main role is to form connections between actin filaments in the sarcomere by binding to actin with one terminus and then dimerizing at the other. It seems a bit odd to refer to it as a sarcolemmal protein.

Response: Thank you for the clarification. We reworded all mentions of the phrase “sarcolemmal protein” as per your suggestion.

Comment #11: Figure 1B – I don't find the significant GI-related diseases or musculoskeletal diseases on the plot? How did you group these? For some reason, I don't have access to Supplemental Table 2, so, perhaps this is where this data is featured. But since you circle back to the musculoskeletal hypothesis, might consider bringing this into the main figure.

Response: Thank you for the suggestion. We colored and labeled the axis text in Figure 1B according to the phenotype group as per your suggestion.

Comment #12: Figure 2 – would be easier to label samples in a way that the reader can easily understand them – e.g. re-label ATAC_D*** to “ATAC-seq Day ***”.

Response: This change was made as per your suggestion.

Reviewer #3

Overall comment: In the manuscript entitled “Genome-wide association and multi-OMIC analyses reveal new mechanisms for Heart failure” the authors performed genome-wide association meta studies of multiple cohorts with heart diseases. The authors replicate the findings in a population cohort. Two loci have been further characterized in terms of their potential functional relevance and possible mechanistic explanations have been provided.

There are conceptual issues with this study that are partly due to the nature of meta-analysis, but also due to the design of their general strategy and functional work-up.

Response: We would like to thank the reviewer their evaluation of our manuscript. We address each point below and seek to clarify some aspects of our design of our study and analyses.

Comment #1: The use of all cardiovascular traits as proxies of heart failure will likely result in higher rates of false negatives and might suggest to the reader that previous findings from original papers cannot be replicated. However, this is due to the assumptions of the authors that every cardiovascular trait is involved in heart failure (either as risk factor, co-morbidity or co-causal factor as shown for afib). As an example of the problematic strategy the reviewer refers to Figure 1B, where you find significant associations between “abdominal hernia” and “ischemic heart failure” or “arrhythmia”. This makes little sense and will not propel the research on heart failure.

Response: We would like to thank the reviewer for this comment because it allows us to clarify a point that may have been misrepresented on the first version of our manuscript. The reviewer seems to have interpreted our manuscript to indicate that we used other cardiovascular diseases as proxies for heart failure in our GWAS meta-analysis. That is not the case, and we seek to clarify any misunderstanding or misrepresentation in our original submission. As detailed in our Response to Reviewer 2, Comment #1, heart failure presence was either explicitly adjudicated by a committee or defined based on relevant ICD codes assigned by physicians in hospitalized patients. Therefore, a participant would be defined as a heart failure case only if they actually have heart failure as determined by the adjudication committee or their physician providers, and not if they have other cardiovascular disorders without heart failure (those participants would in fact be classified as controls). To that end we also included Supplemental Table 11 that lists the definition of heart failure in each of our cohorts.

Of course, it is a fact that most cardiovascular diseases can lead to the development of heart failure (PMID 26935038 and PMID 23747642 are just some examples of the exhaustive literature on the subject). This leads to an enrichment of heart failure cases for other cardiovascular traits and therefore it is not surprising to see not only in our analysis but also in other GWAS on heart failure that some variants associated with other cardiovascular traits (like in our case PITX2 and atrial fibrillation) are showing up as significant. Since heart failure is a multifactorial disease, this is going to be true no matter how strict the definition of disease cases and it has been shown in other similarly multifactorial traits as well (PMID 28135244). However, we would argue that the strength of GWAS in these situations stems from the fact that beyond variants linked to other traits, they are able to identify variants with no other known disease associations, for example the ACTN2 locus in our case, which could point to variants that predispose to disease development regardless of the initial cause.

Lastly, the reviewer points to some associations in Figure 1B to make the case that associations presented in the figure are likely false. We should clarify that Figure 1B presents the genetic correlation analysis. By its nature, genetic correlation cannot be used to infer causality and we have clarified this further in the revision. It merely represents associations between the heritability of different traits which could appear as the result of multiple phenomena, including an actual causal relationship between the compared traits, an enrichment of cases of one trait for the other, shared regulatory mechanisms between relevant cell types, and the consequences of pleiotropy whereby genetic variants have multiple independent effects and influence both traits via independent mechanisms. We agree with the reviewer that the associations we see between abdominal hernia, other GI disorders and skeletal disorders with cardiovascular traits are unlikely to represent causal links between these disease states. However, we would argue that these associations are in fact interesting as they could represent plausible yet underappreciated pleiotropic effects of regulatory mechanisms shared between tissues that have similar cell type composition (muscle, connective tissue and epithelial lineage cells) and could therefore drive both heart muscle disorders and other musculoskeletal diseases. In that sense we maintain that these associations could propel further research on the subject which might reveal interesting shared genetic mechanisms. We would like to highlight seminal papers that describe in much more detail than we ever could in this revision the benefits and limitations of genetic correlation analysis (PMID 31171865 and 30374074) as well as the intriguing and yet unexplained associations noted by others between cardiovascular and musculoskeletal disorders, including abdominal hernia (PMID 25146705 and 28004796).

We revised our manuscript to include some of this discussion and provide clarifications as follows:

Page 4, Paragraph 3: *“We should note that genetic correlation analysis should not be viewed as evidence for a causal relationship between the tested diseases and consequently these results do not indicate that heart failure is causally influenced by musculoskeletal disorders. However, these disorders may share some genetic factors or cellular pathways – since the heart is composed mostly of muscle and stromal tissue, it is plausible that it could share regulatory mechanisms with other organs of similar cell type composition.”*

Comment #2: The p-values for screening and replication are quite modest assuming the cohort sizes and also underline the conceptual problem of this approach.

Response: Thank you for the comment. As discussed in our response to Comment #1 of the reviewer in the text above and in our introduction, we agree that the multifactorial nature of heart failure is challenging and leads to decreased power to detect genetic signals. Nevertheless, and despite this limitation, all our identified genome-wide significant variants meet the Bonferroni adjusted genome wide significance and replication thresholds in our discovery and replication cohorts respectively. This is the standard threshold for reporting GWAS replication (eg. PMID 28715421).

Comment #3: The “multi-omics” analysis is introducing a positive selection bias in its current form and it seems like picking raisins. Both investigations on ABO and ACTN2 are underlined by highly selective strategies using diverse datasets and methods to underline positive associations on genetic, epigenetic and partially in-vitro studies. This part has to be redone completely and performed on the whole genome-wide SNP data, all in the same manner and corrected for multiple testing in this setting.

Response: We respectfully disagree with this comment, and seek to clarify the goal of these targeted analyses. The aim of the presented analyses is not to assess mechanism genome-wide, and in these sections of our manuscript, we do not claim to identify the genome-wide mechanisms behind genetic influence on heart failure. Instead, our approach aims to provide characterization and candidate mechanistic explanations of the individual genome-wide significant associations, and to provide guidance for further research and interpretation of these specific loci. Notably, such “post-GWAS” analysis of significant loci is now considered a critical step in GWAS, being employed for both disease etiology understanding and drug target discovery. This has been highlighted by multiple seminal papers in the field (PMID 26287746, 28753427 and 24952745 to name a few), and the recently published white paper by the International Common Disease Alliance (<https://www.icda.bio/sites/default/files/2019-09/ICDA%20Draft%20White%20Paper.pdf>). Characterization of individual loci has indeed led

to identification of plausible pathways and specific drug targets in some notable cases (eg. PMID 12730697). We have now extended the discussion of the downstream characterization, and added experimental validation, but maintain their utility toward the important goals of characterizing specific loci and supporting future experimental investigation.

Comment #4: The title should be changed completely. The main features are “meta-analysis”, “cardiovascular traits” and “associations”. Everything else is clearly an overstatement.

Response: Although we do think our work points to candidate novel regulatory mechanisms that control the expression of ACTN2 and ABO and lead to heart failure, we acknowledge that the word “mechanism” may indicate a broader claim to some readers, and we have therefore changed the title to a more modest wording according to the reviewer’s request.

Comment #5: More evidence at the protein level and the “real” functional level must be provided for the ACTN2 and ABO loci to strengthen their role in heart failure as suggested in the title.

Response: As detailed in our response to Reviewer 1, Comment #2 above, we provide additional experimental validation of the ACTN2 enhancer effects that we predicted via our multi-omic analysis. Further experimental validation of the pathways via which the ACTN2 and ABO locus variants affect heart failure *in vivo* is of course highly important but is beyond the scope of our current manuscript, as is common for major GWAS studies (eg. PMID 31043756 and 26343387).

Comment #6: In addition, the supplemental material should contain the scripts used for the GWAS, preferably a Python Jupyter notebook for more transparency.

Response: We thank the reviewer for the suggestion, and agree with the importance of reproducible analysis. We provide the scripts used for the GWAS in a publicly available Github repository (https://github.com/marvani88/HF_GWAS). Our analysis is primarily in R and bash, which are widely available, non-proprietary, and used in the community. Note that the underlying data is protected by the individual studies themselves, but is indeed available by application on dbgap. We have provided scripts for preprocessing and further steps once the user has obtained them.

Editor’s comments:

Please be aware that for certain types of new data, including most types of genetic data, journal policy is that deposition in a community-endorsed, public repository is generally mandatory prior to publication. Data submission can be a lengthy process, and we strongly suggest that you begin this well in advance of potential publication to avoid delays later on. Please include a statement about data availability (GWAS summary statistics, omics data) in your point-by-point letter accompanying your revisions.

Please see decision letter above for more detailed information about data requirements and policy. If you are unable to make your data publically available for exceptional reasons, please get in touch with me now to discuss this further.

Response: We agree that data sharing is important in the genetics community and we therefore plan to make all data used in our study publicly available as per the journal's policy. Specifically, our GWAS summary statistics will be made available in a public Zenodo repository upon manuscript acceptance, and all the genotype data used to generate those summary statistics are available from dbGaP or via a request to the UK BioBank as detailed in our paper. Our analysis of eQTL data from GTEx and eQTLGen are all available in our supplemental Tables and the corresponding GTEx v8 sequencing data are available from dbGaP and on the GTEx project portal (<https://gtexportal.org/home/>). The epigenetic, RNA-seq and HiC data in hESC-CMs for the loci analyzed in our manuscript will also be made available at the same Zenodo repository upon manuscript acceptance. Lastly, all the CRISPR data on cardiomyocytes are provided in our Supplemental Tables. We added a data availability section in our manuscript to reflect the above:

Page 17 Paragraph 1: *“Our GWAS summary statistics were made available in a public Zenodo repository (can be accessed via Github on https://github.com/marvani88/HF_GWAS), and all the genotype data used to generate those summary statistics are available from dbGaP (accession numbers phs000007.v29.p11, phs000287.v6.p1, phs000209.v13.p3, phs000280.v4.p1, phs000200.v11.p3, phs000888.v1.p1) or via a request to the UK BioBank. Our analysis of eQTL data from GTEx and eQTLGen are all available in our Supplemental Tables and the corresponding GTEx v8 sequencing data are available from dbGaP (accession number phs000424.v8.p2) and on the GTEx project portal (<https://gtexportal.org/home/>). Sequencing data from the cardiomyocyte differentiation experiments have been deposited in the Gene Expression Omnibus under the accession number GSE116862 whereas the processed epigenetic, RNA-seq and HiC data in hESC-CMs for our loci of interest were also made available at the following Github repository (https://github.com/marvani88/HF_GWAS). Lastly, all the CRISPR data on cardiomyocytes are provided in our Supplemental Tables.”*

REVIEWERS' COMMENTS:

Reviewer #1 (Remarks to the Author):

The authors have fully addressed my comments. I agree that deletion of the ACTN2 enhancer is adequate to demonstrate a functional relationship between this non-coding locus and gene expression. Single variant editing is not necessary for the claims they have made in the manuscript

Reviewer #2 (Remarks to the Author):

"The authors have answered all of my initial queries in a satisfactory manner. They have invested significant effort into developing mechanistic evidence for their observations, better defining relevant methods and study populations, and bolstering their findings with existing data more broadly. Overall I think this is a carefully designed, well-executed and judiciously interpreted study, and represents an important addition to the heart failure literature."

Reviewer #3 (Remarks to the Author):

In the revised version of the manuscript "Genome-wide association and multi-omic analyses reveal new insights for Heart Failure" the authors addressed some of the concerns of this reviewer, especially in clarifying that all individuals suffered from heart failure by providing Suppl. Table 11. Also, they revised the title to better address their main findings and avoid overinterpretation of this meta-GWAS.

This reviewer is, however, still skeptical about principle aspects of this work that may be prone for over- or misinterpretation. The functional investigations are very selective and in the revision there is a plethora of examples on the limited mechanistic proof. The authors argue that such strategies have yielded important discoveries, which is true. However, such major discoveries are certainly not made in this study. As an example may serve the link between rs9411378 and ABO. The given publications link the lead SNP and blood count, metabolic disorders and venous thromboembolism. The conditioning analysis reveals independence of HF risk factors. When excluding CAD, which is known strong predictor of HF, the association is not significant anymore (given p-value $1,3 \times 10^{-4}$; although claimed that association persists). The functional evidence is very poor and only speculative nature. There is no active enhancer or promoter in myocardial tissue, but only in datasets from hematopoietic cells. The splicing association is also referring to blood data from public datasets. This leaves any functional insights elusive and there is not a single experiment establishing the mechanistic link between lead SNP, ABO and HF. The final conclusion "...link to development of heart failure" again aims to render these findings mechanistic, but no given data underline a link to the "development" of HF. The reviewer does not think that this line of evidence furthers the dissection of heart failure on a mechanistic basis.

Response to Reviewers

We would like to sincerely thank the referees for their thorough review of our manuscript. We incorporated their suggestions and provide point-by-point responses below.

Reviewer #1 (Remarks to the Author):

The authors have fully addressed my comments. I agree that deletion of the ACTN2 enhancer is adequate to demonstrate a functional relationship between this non-coding locus and gene expression. Single variant editing is not necessary for the claims they have made in the manuscript

We would like to thank the reviewer for their time and effort in reviewing our manuscript. Their comments were thoughtful and led to an improved manuscript.

Reviewer #2 (Remarks to the Author):

"The authors have answered all of my initial queries in a satisfactory manner. They have invested significant effort into developing mechanistic evidence for their observations, better defining relevant methods and study populations, and bolstering their findings with existing data more broadly. Overall I think this is a carefully designed, well-executed and judiciously interpreted study, and represents an important addition to the heart failure literature."

We would like to thank the reviewer for their time and effort in reviewing our manuscript. Their comments were thoughtful and led to an improved manuscript.

Reviewer #3 (Remarks to the Author):

In the revised version of the manuscript "Genome-wide association and multi-omic analyses reveal new insights for Heart Failure" the authors addressed some of the concerns of this reviewer, especially in clarifying that all individuals suffered from heart failure by providing Suppl. Table 11. Also, they revised the title to better address their main findings and avoid overinterpretation of this meta-GWAS.

This reviewer is, however, still skeptical about principle aspects of this work that may be prone for over- or misinterpretation. The functional investigations are very selective and in the revision there is a plethora of examples on the limited mechanistic proof. The authors argue that such strategies have yielded important discoveries, which is true. However, such major discoveries are certainly not made in this study. As an example may serve the link between rs9411378 and ABO. The given publications link the lead SNP and blood count, metabolic disorders and venous thromboembolism. The conditioning analysis reveals independence of HF risk factors. When excluding CAD, which is known strong predictor of HF, the association is not significant anymore (given p-value 1.3×10^{-4} ; although claimed that association persists). The functional evidence is very poor and only speculative nature. There is no active enhancer or promoter in myocardial tissue, but only in datasets from hematopoietic cells. The splicing association is also referring to blood data from public datasets. This leaves any functional insights elusive and there is not a single experiment establishing the mechanistic link between lead SNP, ABO and HF.

The final conclusion "...link to development of heart failure" again aims to render these findings mechanistic, but no given data underline a link to the "development" of HF. The reviewer does not think that this line of evidence furthers the dissection of heart failure on a mechanistic basis.

We would like to thank the reviewer for their time and effort in reviewing our manuscript. We agree that we have not provided mechanistic evidence for the relationship between rs9411378 and heart failure but we believe that such a mechanistic link, although important, is beyond the scope of this work. However, we strongly believe that the evidence for association between this chromosome 9 locus and heart failure is indisputable as the locus is highly significant in both the discovery and replication analysis. We therefore strongly disagree with the reviewer that this line of evidence does not further the dissection of heart failure on a mechanistic basis as we believe that this data do in fact serve as hypothesis-generating for future studies that delve deeper into the mechanistic basis of this association.

We believe that the sensitivity analysis in question which was performed per the request of reviewer #1 in conjunction with our multi-trait conditional analysis conditioning on the effect of CAD do lend additional support to the notion that the discovered association between rs9411378 and heart failure cannot be fully explained by its association with CAD. That said, we concur with the reviewer that since the analysis excluding CAD is underpowered and the p-value does not reach statistical significance it cannot be construed as definitive evidence of an independent effect. We have therefore revised the wording in our manuscript to provide some additional clarity regarding these nuances.

The revised text reads:

"In addition, since ABO is a known locus for coronary disease, which in turn is one of the major disorders leading to heart failure, beyond conditioning on ischemic heart disease we also performed a sensitivity analysis in which we excluded all patients with coronary artery disease (CAD) and tested the association between the locus sentinel SNP and heart failure ($\log(\text{Odds Ratio})=0.1027$, score test $p\text{-value}=1.3e-4$). Since CAD is only one of the many causes of heart failure, and individuals can have both CAD and heart failure from a different cause, the restricted analysis is conservative and consequently underpowered compared to our discovery GWAS ($N_{\text{cases}}=4,137$ vs $N_{\text{cases}}=10,976$), which even at the expected, unrestricted effect size inevitably makes the association non-significant at a genome-wide $p\text{-value}$ threshold of $5e-8$. Nevertheless, the restricted effect size on HF did remain similar to our unrestricted analysis and the association remained nominally significant. Taken together, this sensitivity analysis and the multi-trait conditional analysis suggest the possibility of a role of the ABO locus on heart failure independent of its established influence on coronary disease risk. However, definitive proof of this hypothesis will require further study."